# PR-SET7 epigenetically restrains uterine interferon response and cell death governing proper postnatal stromal development

Haili Bao[1,3], Yang Sun[1,3], Na Deng[1,3], Leilei Zhang[1], Yuanyuan Jia[1], Gaizhen Li[1], Yun Gao[1], Xinyi Li[1], Yedong Tang[1], Han Cai[1], Jinhua Lu ®[1], Haibin Wang ®[1,2] ✉, Wenbo Deng ®[1] ✉ & Shuangbo Kong ®[1] ✉

The differentiation of the stroma is a hallmark event during postnatal uterine development. However, the spatiotemporal changes that occur during this process and the underlying regulatory mechanisms remain elusive. Here, we comprehensively delineated the dynamic development of the neonatal uterus at single-cell resolution and characterized two distinct stromal subpopulations, inner and outer stroma. Furthermore, single-cell RNA sequencing revealed that uterine ablation of *Pr-set7*, the sole methyltransferase catalyzing H4K20me1, led to a reduced proportion of the inner stroma due to massive cell death, thus impeding uterine development. By combining RNA sequencing and epigenetic profiling of H4K20me1, we demonstrated that PR-SET7-H4K20me1 either directly repressed the transcription of interferon stimulated genes or indirectly restricted the interferon response via silencing endogenous retroviruses. Declined H4K20me1 level caused viral mimicry responses and ZBP1-mediated apoptosis and necroptosis in stromal cells. Collectively, our study provides insight into the epigenetic machinery governing postnatal uterine stromal development mediated by PR-SET7.

The uterus, which is derived from the Müllerian duct, serves as an essential organ for reproduction in mammals. Disturbances during uterine development exert ripple effects on uterine structure and functions in adulthood and give rise to infertility[1].

In mice, the uterus is relatively rudimentary at birth (postnatal day (PND) 1), consisting of a single layer of luminal epithelium surrounded by undifferentiated mesenchyme. The emergence of endometrial glands through invagination and branching from PND5 to PND12, as well as the coordinated differentiation of the mesenchyme into endometrial stroma and myometrium, is considered two milestone events that occur during postnatal uterine development in rodents. In mice, the fundamental configuration of the uterus is established by PND15[2–4]. While a variety of studies have identified many crucial genes

that participate in the formation of uterine glands[5–9], the dynamic process and underlying regulatory mechanisms of postnatal uterine stromal development remain largely elusive. A previous report proposed that a subluminal population of *Misr2* (Müllerian inhibiting substance receptor 2)-positive cells functioned as progenitor cells, which further differentiated into two stromal subclusters according to single-cell RNA sequencing (scRNA-seq)[10]. However, the scRNA-seq analysis was performed at only one time point in this study and was incapable of providing a comprehensive landscape of stromal development.

Epigenetic machinery, including DNA methylation, histone modifications, and other modifications affecting chromatin accessibility and three-dimensional chromatin organization, plays a vital role in

[1]Fujian Provincial Key Laboratory of Reproductive Health Research, Department of Obstetrics and Gynecology, The First Affiliated Hospital of Xiamen University, School of Medicine, Xiamen University, Xiamen, Fujian 361102, China. [2]State Key Laboratory of Vaccines for Infectious Diseases, Xiang An Biomedicine Laboratory, School of Medicine, Xiamen University, Xiamen, Fujian 361102, China. [3]These authors contributed equally: Haili Bao, Yang Sun, Na Deng. ✉e-mail: haibin.wang@vip.163.com; wbdeng@xmu.edu.cn; shuangbo_kong@163.com

development and disease via controlling gene expression[11]. PR-SET7, an evolutionarily conserved histone methyltransferase, is the sole enzyme known to catalyze H4K20me1, which serves as the substrate for H4K20me2/3[12,13]. In addition to its well-accepted canonical functions in cell cycle progression, DNA damage repair, and gene transcriptional regulation[14–19], PR-SET7-mediated H4K20me1 was recently demonstrated to repress the intrinsic expression of endogenous retroviruses (ERVs), contributing to the maintenance of genomic stability[20]. Accumulating evidence has uncovered the indispensable roles of PR-SET7 and H4K20me1 in diverse physiological and pathological processes during the development and homeostasis of multiple organs and tissues[21–25].

The pathophysiological significance of PR-SET7-mediated H4K20me1 in female reproduction has been reported. Trophoblast-specific deletion of *Pr-set7* resulted in ERV derepression, viral mimicry responses, and necroptosis, which shed light on the potential etiology and pathogenesis of recurrent miscarriage[20]. Meanwhile, uterine PR-SET7 deficiency limited uterine epithelial population growth due to impaired DNA damage repair and severe epithelial cell apoptosis, which hampered gland formation in neonatal mice[26]. Nevertheless, the potential contributions of PR-SET7-mediated H4K20me1 to uterine stromal development have been neglected and remain ambiguous.

In the current study, we generate a dynamic transcriptomic atlas of uterine development in mice throughout the first two weeks after birth using scRNA-seq and describe the spatial localization, developmental trajectory, and unique characteristics of two stromal subsets, inner and outer stromal cells, in the neonatal uterus. In addition, we discover that uterine PR-SET7 deficiency disrupts the development of inner stromal cells and results in considerable cell death, leading to uterine atrophy. Utilizing a combination of bulk RNA sequencing (RNA-seq) and cleavage under targets and tagmentation (CUT&Tag) approaches, we elucidate that PR-SET7-mediated H4K20me1 epigenetically confines the aberrant activation of interferon responses and prevents ZBP1-mediated apoptosis and necroptosis in stromal cells, which sheds light on the pivotal role of PR-SET7-mediated H4K20me1 in calibrating interferon signaling to safeguard proper development of the endometrial stroma.

## Results

### The dynamic transcriptomic landscape of postnatal uterine development unraveled by scRNA-seq

The highly heterogeneous uterus is composed of various cell types and undergoes tremendous changes during development[4,9,10]. To achieve a comprehensive understanding of the dynamic process of postnatal uterine development, uterine tissues were collected on PND1, PND5, PND10 and PND15 and subjected to scRNA-seq (Fig. 1a). After quality control and batch effect correction, 13 distinct cell clusters were identified as visualized by uniform manifold approximation and projection (UMAP), including three stromal (Str) subpopulations, three epithelial (Epi) subpopulations, two myometrial (Myo) subpopulations, pericytes (Peri), mesothelial (Meso) cells, endothelial (Endo) cells, T cells and macrophages (Mac), according to well-documented marker genes (Fig. 1b–d and Supplementary Fig. 1a, b). The proportions of stromal, epithelial, and myometrial cells fluctuated wavily. Meanwhile, the percentages of pericytes, endothelial cells, and leukocytes increased gradually throughout the postnatal period (Fig. 1e), reflecting the maturation of the uterine vasculature and the establishment of a local immune microenvironment.

Stromal cells were the most abundant cell type in the neonatal uterus (Fig. 1b, e), consisting of Str_1 (which specifically expressed Wnt family member 16, *Wnt16*), Str_2 (which was characterized by relatively high level of HtrA serine peptidase 3, *Htra3*) and Str_3, a proliferative subset (Fig. 1d). The proportion of proliferative stromal cells gradually decreased, substantially dropping by PND15 (Fig. 1f), which coincided with the accomplishment of uterine development. In subsequent

studies, we mainly put emphasis on Str_1 and Str_2. Both Str_1 and Str_2 already existed at birth. They seemed less differentiated since the expression levels of their marker genes were nearly undetectable at this time point. In the next two weeks, the expression levels of the marker genes of both Str_1 and Str_2 increased, and the expression pattern became more restricted (Fig. 1i, j), indicating the differentiation of stromal subpopulations.

The three epithelial subpopulations were defined as the luminal epithelial cells (Epi_1), a group of proliferative epithelial cells (Epi_2), and the glandular epithelial cells (Epi_3), based on the expression of tumor-associated calcium signal transducer 2 (*Tacstd2*), marker of proliferation Ki67 (*Mki67*) and forkhead box A2 (*Foxa2*), respectively (Fig. 1d). The formation of endometrial glands from PND5 to PND12 is a milestone event during uterine development[3]. Consistently, *Foxa2*-positive glandular epithelial cells appeared on PND10, and the number of these cells was significantly increased on PND15 (Fig. 1g, k).

The actin alpha 2 (*Acta2*)-positive uterine myometrium was subdivided into non-proliferative (Myo_1) and proliferative (Myo_2) myometrial cells, depending on *Mki67* expression (Fig. 1d). The proportion of proliferative myometrial cells tended to increase from PND1 to PND10 (Fig. 1h), aligned with the progression of uterine myometrial development[2]. We also noticed myometrial cells with low expression level of *Acta2* on PND1 before the morphological appearance of the myometrium[2] (Fig. 1l), implying that the fates of these cells had already been determined at birth, and they remained in a less-differentiated state until the emergence of the myometrial layers.

Collectively, we depicted the dynamic process of uterine development after birth at single-cell resolution and demonstrated that although the uterine mesenchyme seemed homogeneous at birth, the less-differentiated stromal and myometrial cells had already undergone cell fate determination, earlier than previously thought.

### Characterization of distinct stromal subpopulations in the neonatal uterus

To elucidate the localization of Str_1 and Str_2, single-cell resolution in situ hybridization on tissues (SCRINSHOT) was performed using specific probes for their respective marker genes, *Wnt16* and *Htra3*. *Wnt16*-positive Str_1 was mainly distributed adjacent to the luminal epithelium, and this subpopulation was defined as the inner stromal cells (Fig. 2a). Meanwhile, Str_2, which highly expressed *Htra3*, was localized at the periphery of the endometrium, and these cells were defined as the outer stromal cells (Fig. 2b).

The inner and outer stroma displayed prominent differences in global gene expression (Supplementary Fig. 2a). To systematically decode the developmental trajectory of these stromal subclusters, we applied RNA velocity analysis and observed a potential differentiation orientation from outer stromal cells to inner stromal cells (Fig. 2c). In addition, single-cell regulatory network inference and clustering (SCENIC) analysis was performed to decipher the potential transcription factors governing the differentiation of inner stromal cells (Fig. 2d and Supplementary Fig. 2b). We found that the inner stromal cells gradually acquired progesterone receptor (PR) expression, which remained at a relatively low level in the outer stromal cells (Fig. 2e–g), suggesting the capability of inner stromal cells to better respond to progesterone in the adulthood.

To better depict the unique characteristics of the inner and outer stromal cells, genes that were highly and specifically expressed in these two subsets were selected and subjected to the gene ontology (GO) enrichment analysis. The outer stromal cells were involved in pathways related to extracellular matrix structure and collagen fibril organization (Fig. 2h), consistent with the RNA velocity result that this stromal subpopulation exhibited a less-differentiated state. On the other hand, the inner stromal cells particularly expressed genes relevant to WNT signaling, BMP signaling as well as mesenchyme and gland development (Fig. 2h), implying their potential roles in regulating the

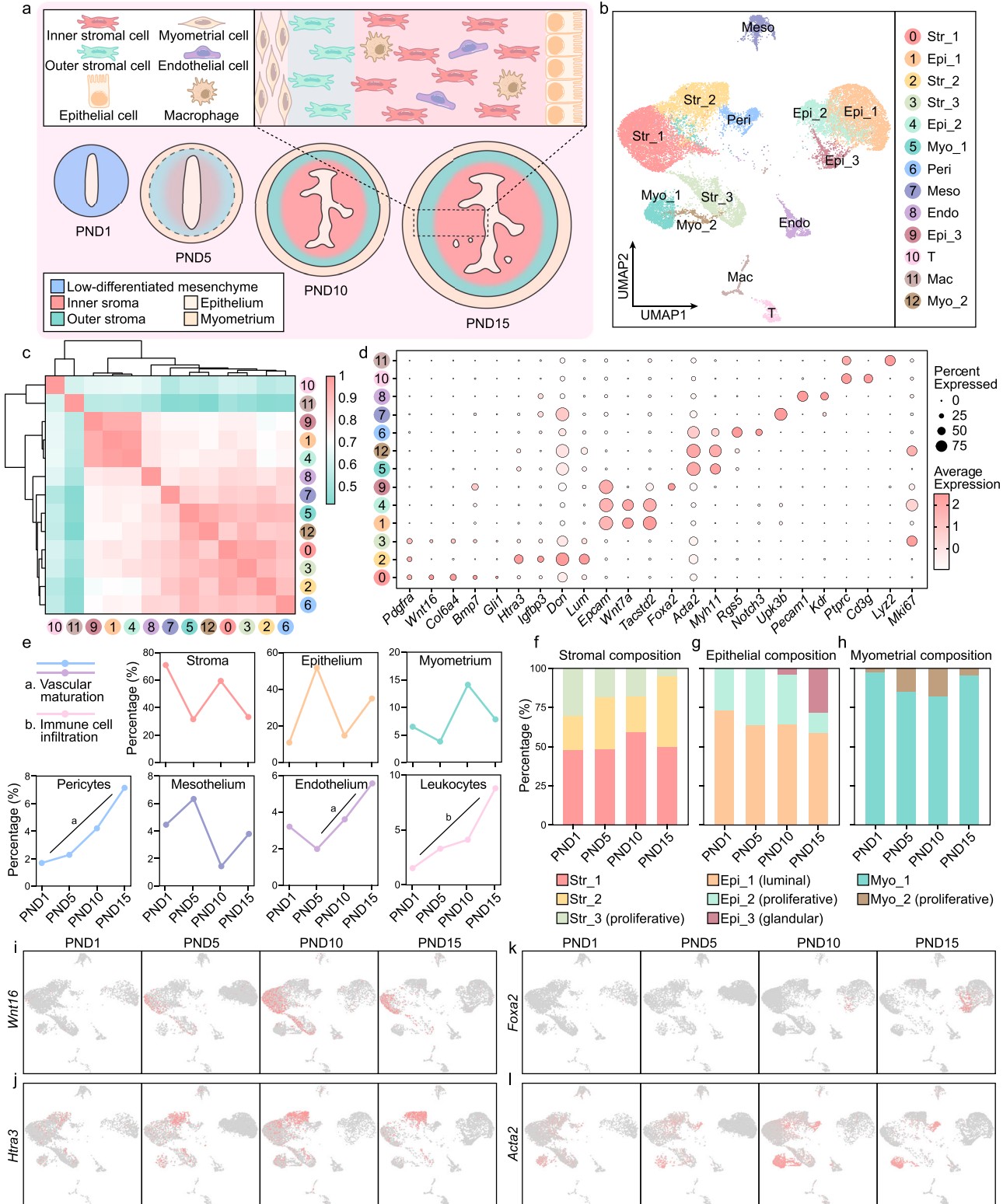

**Fig. 1 | The dynamic transcriptomic atlas of the postnatal uterus depicted by scRNA-seq. a** Diagram illustrating the dynamic process of postnatal uterine development. **b** UMAP plot showing cell clusters in the neonatal uterus. **c** Spearman correlation of transcriptomic profiles of different cell populations. **d** Bubble plot indicating the expression of marker genes across different cell types. **e** Line chart showing the proportions of the major cell types at different time points. **f–h** Bar plot showing the stromal (**f**), epithelial (**g**), and myometrial (**h**) subset composition on PND1, PND5, PND10, and PND15. **i–l** UMAP visualization indicating the expression of *Wnt16* (**i**), *Htra3* (**j**), *Foxa2* (**k**), and *Acta2* (**l**) at different time points.

formation of uterine glands, as previously reported[27]. We noticed that the inner stromal cells highly expressed genes involved in chromatin remodeling and RNA processing on PND1 and PND5. On PND10, pathways related to WNT signaling and epithelial tube morphogenesis showed enrichment (Fig. 2i), synchronized with the emergence of uterine glands (Fig. 1g). Consistently, CellChat analysis revealed intimate interactions between inner stromal cells and glandular epithelial cells mediated by the WNT and BMP signaling pathways (Fig. 2j, k and

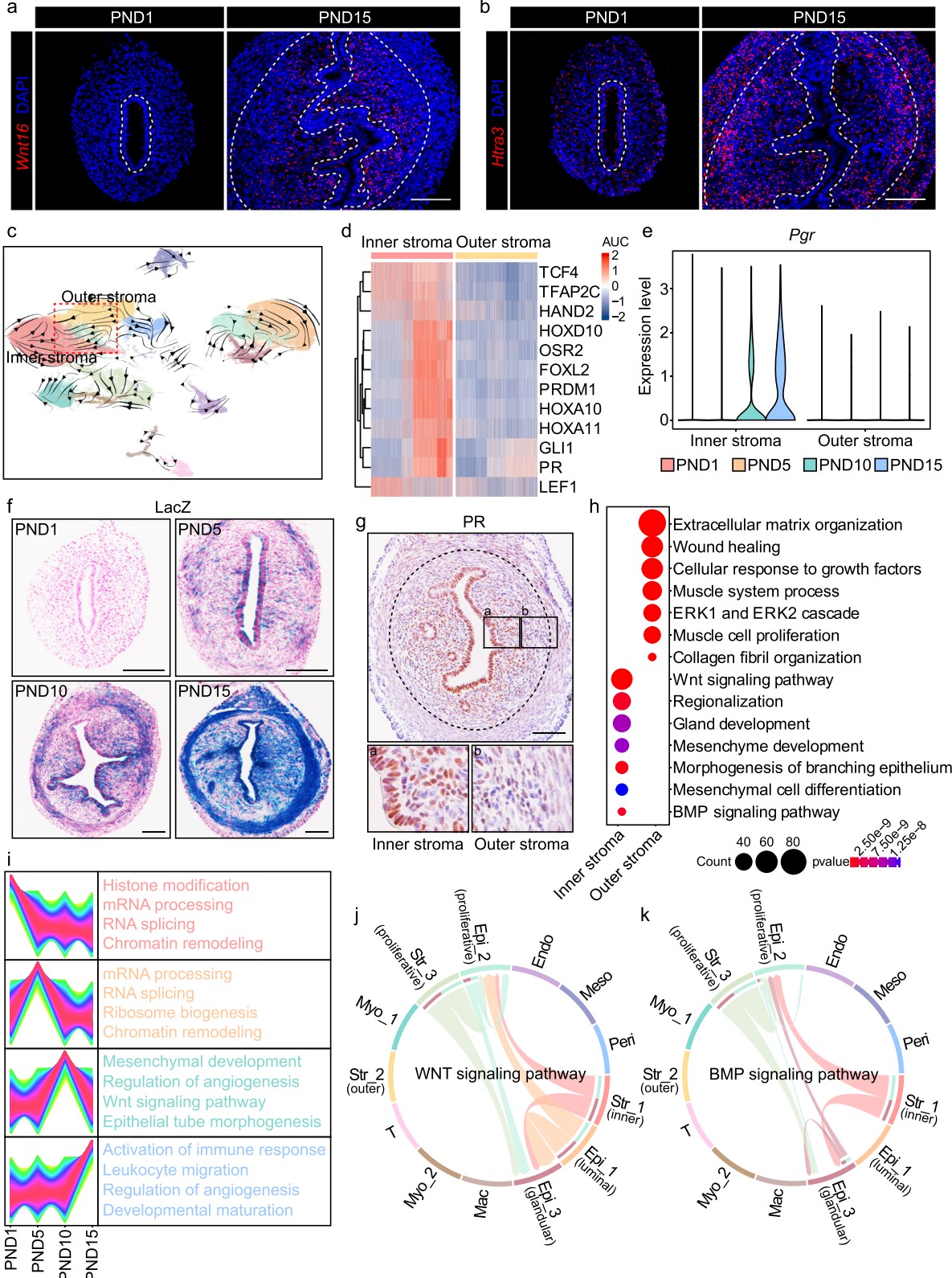

Supplementary Fig. 2c). The expression of genes related to angiogenesis and leukocyte chemotaxis was significantly increased on PND10 and PND15, revealing the role of inner stromal cells in promoting these critical events (Fig. 2i and Supplementary Fig. 2d).

Overall, our scRNA-seq data provided detailed insights into the spatial localization, developmental trajectory, and unique features of different stromal subsets.

**Uterine-specific deletion of *Pr-set7* hampered stromal growth due to massive cell death**

Our previous study focused mainly on the contribution of PR-SET7 to adenogenesis; however, we noticed that endometrial stromal growth was also impeded in the absence of PR-SET7[26]. Therefore, in the present study, we intended to elucidate the physiological significance of PR-SET7-mediated H4K20me1 in uterine stromal development. We

**Fig. 2 | The descriptions of different stromal subsets in the neonatal uterus.** **a**, **b** SCRINSHOT analysis of *Wnt16* (**a**) and *Htra3* (**b**) in the uteri on PND1 and PND15. Dash lines represent the boundary between the epithelium and the stroma, as well as the stroma and the myometrium. Scale bar: 100 μm. **c** Visualization of RNA velocity analysis. **d** Heatmap showing the top-ranked regulons in inner stromal cells according to SCENIC analysis. **e** Violin plot displaying the expression level of *Pgr* in the inner and outer stroma throughout postnatal uterine development. **f** LacZ staining showing the expression pattern of *Pgr* in the uteri on PND1, PND5, PND10, and PND15. Scale bar: 100 μm. **g** IHC analysis of PR in the uteri on PND15. Dash lines represent the boundary between the stroma and the myometrium. Scale bar: 100 μm. **h** GO enrichment analysis of the highly expressed genes in the inner and outer stroma. Significance is based on over-representation analysis with cluster-Profiler. **i** Gene expression clustering analysis and GO analysis of inner stromal cells at different time points. **j**, **k** CellChat analysis indicating the WNT (**j**) and BMP (**k**) signaling pathways among stromal cells and epithelial cells.

first assessed the spatiotemporal patterns of *Pr-set7* expression and H4K20me1 modification in the neonatal uterus using in situ hybridization (ISH), immunofluorescence (IF) and western blotting (WB) (Fig. 3a–c). *Pr-set7* and H4K20me1 signals were detected in all uterine cell types throughout the first two weeks after birth (Fig. 3a, b). The existence of H4K20me1 in the stromal compartment was further confirmed by co-staining with the stromal cell marker Wilms tumor gene 1 (WT1) (Fig. 3d).

In order to unravel the potential role of PR-SET7 in uterine development after birth, mice with uterine-specific ablation of *Pr-set7* (hereafter referred to as *Pr-set7*[d/d]) were generated by crossing *Pr-set7 floxed* mice (hereafter referred to as *Prset7*[f/f]) with *Pgr-IRES-Cre* mice. Specifically, the 7th exon of *Pr-set7*, which encodes the SET domain that is responsible for its methyltransferase activity, was deleted. According to LacZ staining, the Cre recombinase exhibited activity by PND5 in the uterus (Fig. 2f). On PND5, the *Pr-set7* mRNA was efficiently deleted in the uterus according to quantitative reverse transcription-polymerase chain reaction (qRT-PCR) (Fig. 3e). Furthermore, the level of H4K20me1 was obviously declined upon *Pr-set7* deletion, as revealed by WB (Fig. 3f) and IF (Fig. 3g).

Given that the murine uterus develops within two weeks after birth, we monitored the growth of the endometrial stroma throughout this pivotal period. As shown by IF staining of VIMENTIN and platelet-derived growth factor receptor alpha (PDGFRα), the area of the uterine stroma increased to more than three times its original size from PND5 to PND15 in *Pr-set7*[f/f] females. In contrast, the stromal area in the *Pr-set7*[d/d] group was even reduced (Fig. 3h–j).

It has been widely reported that PR-SET7 serves as a multifunctional switch for cell cycle and cell survival[14]. We thus surmised that the hampered development of the endometrial stroma in the *Pr-set7*[d/d] uterus resulted from aberrant cell proliferation or death. We examined the expression of several cyclins and cyclin-dependent kinase inhibitors (CKIs), among which p21 and p16 were markedly upregulated due to *Pr-set7* deletion (Supplementary Fig. 3a, b). Nevertheless, the upregulated p21 was only detected in epithelial cells (Supplementary Fig. 3c), implying that the cell cycle progression was less affected in the *Pr-set7*[d/d] stroma. Consistently, the proliferation of stromal cells in the *Pr-set7*[d/d] uterus rivaled that in the *Pr-set7*[f/f] group, as visualized by immunohistochemistry (IHC) analysis of Ki67, proliferating cell nuclear antigen (PCNA) and phospho-histone H3 (p-H3) (Supplementary Fig. 3d–f). Subsequently, we evaluated stromal cell death using terminal deoxynucleotidyl transferase-mediated dUTP nick-end labeling (TUNEL). Barely any stromal cells in the *Pr-set7*[f/f] uterus exhibited TUNEL signals, while a large portion of the *Pr-set7*[d/d] stromal cells underwent cell death (Fig. 3k, l).

These findings proved that the limited growth of the endometrial stroma upon PR-SET7 loss was due to massive stromal cell death rather than impaired proliferation.

**Uterine PR-SET7 deficiency led to a reduced number and derailed differentiation of inner stromal cells**

To further delineate defective uterine stromal development due to *Pr-set7* ablation, we compared stromal cell composition and gene expression between the *Pr-set7*[f/f] and *Pr-set7*[d/d] groups using scRNA-seq (Fig. 4a, b). An exaggerated decrease in the proportion of epithelial cells, especially glandular (Epi_3) epithelial cells, was observed in the *Pr-set7*[d/d] group (Fig. 4b), consistent with the previous study[26]. In addition, the proportion of inner stromal cells, which accounted for approximately 60% of total stromal cells in the *Pr-set7*[f/f] uterus, was reduced prominently to 40% upon PR-SET7 loss (Fig. 4b, c).

Consistently, we observed significantly fewer *Wnt16*-positive inner stromal cells and more outer stromal cells that expressed *Htra3* in the *Pr-set7*[d/d] uterus (Fig. 4d). The decline in the number of inner stromal cells was further verified by PR staining (Fig. 4e) since PR was expressed at a higher level in the inner stromal cells (Fig. 2e–g). Moreover, we unexpectedly discovered that PR-positive inner stromal cells vanished entirely in the uteri of 4-week-old *Pr-set7*[d/d] mice (Fig. 4f). Since more dead cells were observed underneath the luminal epithelium in the inner stroma in the absence of PR-SET7 (Fig. 3k), it was rational to speculate that the aberrantly decreased inner stromal subpopulation in the *Pr-set7*[d/d] uterus was ascribed to massive cell death, which led to the complete disappearance of inner stromal cells before puberty.

In addition to the altered stromal subset composition, the transcriptional profiles of both inner and outer stromal cells displayed obvious differences between the *Pr-set7*[f/f] and *Pr-set7*[d/d] groups (Fig. 4g). Genes governing inner stromal differentiation and reproductive structure development, such as the transcription factors *Prdm1* and *Gli1*, were markedly downregulated in the inner stromal cells compared to the outer stromal cells upon *Pr-set7* deletion (Fig. 4h, i). Moreover, the pro-angiogenetic potential of inner stromal cells was also clearly suppressed (Fig. 4h), as visualized by downregulated expression of vascular endothelial growth factor a (*Vegfa*) and adrenomedullin (*Adm*) (Supplementary Fig. 4a). Meanwhile, the upregulated genes in the *Pr-set7*[d/d] inner stroma were particularly enriched in viral defense reactions and innate immune responses (Fig. 4h). These results suggested that uterine deletion of *Pr-set7* interfered with the differentiation and homeostasis of inner stromal cells.

Postnatal uterine development relies on complex communications between the epithelium and the stroma[4]. Accompanied with the downregulation of *Gli1* in the *Pr-set7*[d/d] inner stromal cells, the expression of *Ihh*, the upstream ligand of the Hedgehog signaling, vanished entirely in the epithelium (Fig. 4j). Furthermore, CellChat was applied to dissect out the complex interactions between different cell types in the *Pr-set7*[f/f] and *Pr-set7*[d/d] neonatal uteri. Both the number and the strength of cell-cell interactions were dramatically decreased in the *Pr-set7*[d/d] uterus (Supplementary Fig. 4b-c). The WNT and BMP signaling pathways, which are potentially implicated in gland formation (Fig. 2j, k), were completely abrogated between the inner stroma and the epithelium in the *Pr-set7*[d/d] uterus (Supplementary Fig. 4c). Stromal derived WNT5A, which activates the non-canonical WNT (ncWNT) signaling, has been reported to participate in the formation of glands[27]. We indeed observed a compromised ncWNT signaling pathway between inner stromal cells and epithelial cells in the *Pr-set7*[d/d] group (Fig. 4k).

Collectively, the deficiency of PR-SET7 resulted in a reduced number and compromised differentiation of inner stromal cells, which impaired proper epithelial-stromal interactions during uterine development.

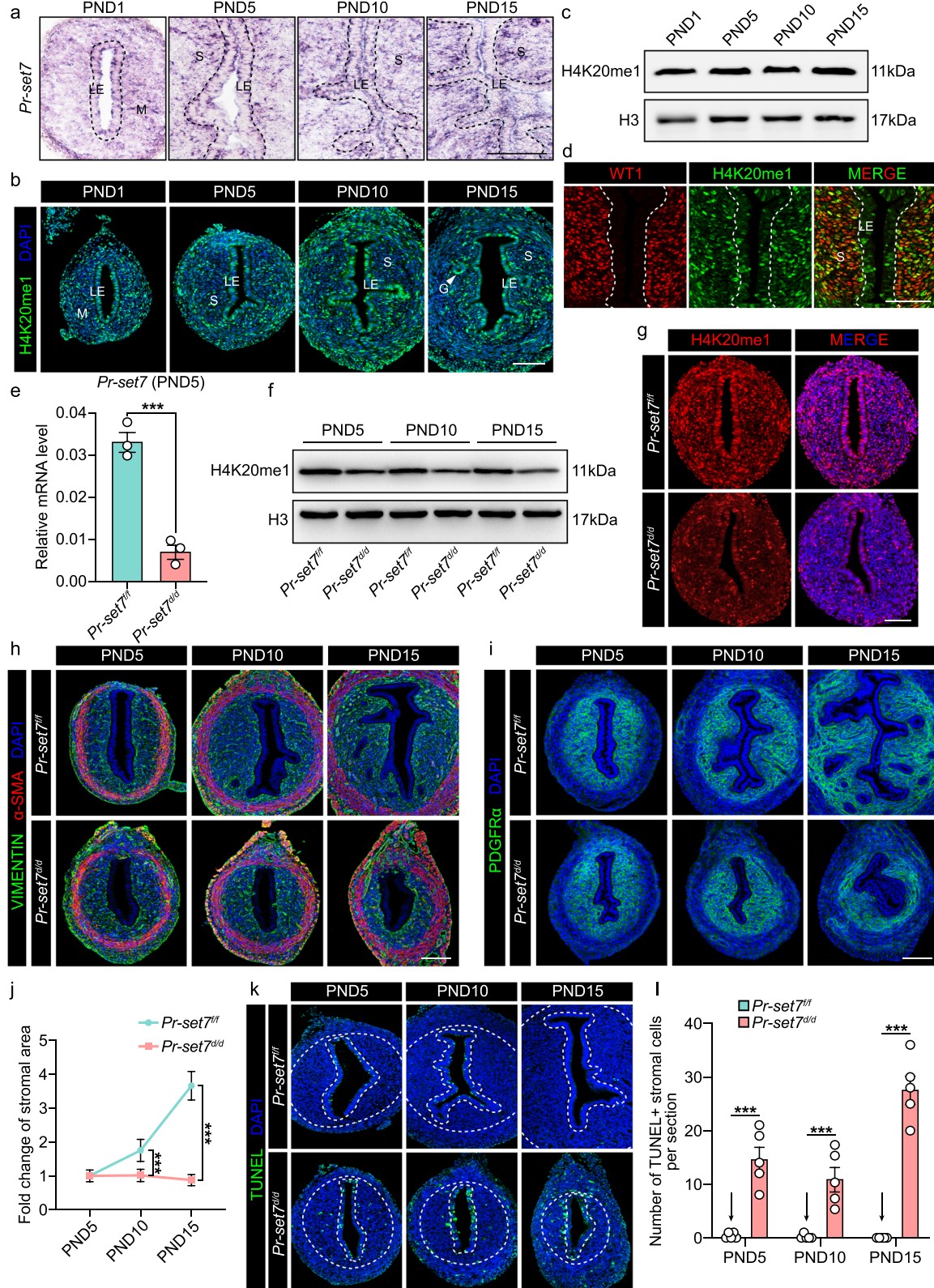

## Interferon responses were aberrantly elicited in the uterus due to PR-SET7 loss

We noticed that many of the upregulated genes in *Pr-set7*^d/d inner stromal cells were enriched in immune activation and apoptotic signaling (Fig. 4h); thus, we conjectured that the massive cell death in the *Pr-set7*^d/d inner stroma was related to aberrantly activated immune responses. Indeed, the expression of interferon stimulated genes (ISGs) was significantly increased in the inner stroma of the *Pr-set7*^d/d uterus (Fig. 5a and Supplementary Fig. 5a). Moreover, CellChat analysis illustrated that several immunity-associated signaling pathways were remarkably activated upon *Pr-set7* deletion (Supplementary Fig. 5b).

To further confirm these findings, the uteri of *Pr-set7*^f/f and *Pr-set7*^d/d mice were subjected to whole transcriptome sequencing. A total of 1347 and 2366 differentially expressed genes (fold change >=1.5,

**Fig. 3 | Uterine-specific ablation of *Pr-set7* resulted in hampered stromal growth due to massive cell death. a** ISH analysis of *Pr-set7* in the uteri on PND1, PND5, PND10, and PND15. Dash lines represent the boundary between the epithelium and the stroma. LE luminal epithelium, M mesenchyme, S stroma. Scale bar: 100 μm. **b** IF analysis of H4K20me1 in the uteri on PND1, PND5, PND10, and PND15. LE luminal epithelium, M mesenchyme, S stroma, GE glandular epithelium. Scale bar: 100 μm. **c** WB analysis of H4K20me1 in the uteri on PND1, PND5, PND10, and PND15. H3 served as a loading control. **d** Co-staining of H4K20me1 and WT1 in the uterus. LE luminal epithelium, S stroma. Dash lines represent the boundary between the epithelium and the stroma. Scale bar: 100 μm. **e** QRT-PCR analysis of *Pr-set7* mRNA level in *Pr-set7^f/f* (n = 3 mice) and *Pr-set7^d/d* (n = 3 mice) uteri on PND5. The values were normalized to the *Gapdh* level. Data are presented as mean ± SEM. Two-tailed unpaired Student's *t*-test. ***p = 0.0008. **f** WB analysis of H4K20me1 in *Pr-set7^f/f* and *Pr-set7^d/d* uteri on PND5, PND10 and PND15. H3 was used as a loading control. **g** IF analysis of H4K20me1 in *Pr-set7^f/f* and *Pr-set7^d/d* uteri on PND5. Scale bar: 100 μm. **h** IF analysis of VIMENTIN and α-SMA in *Pr-set7^f/f* and *Pr-set7^d/d* uteri on PND5, PND10 and PND15. Scale bar: 100 μm. **i** IF analysis of PDGFRα in *Pr-set7^f/f* and *Pr-set7^d/d* uteri on PND5, PND10 and PND15. Scale bar: 100 μm. **j** Fold change of stromal area in *Pr-set7^f/f* (n = 5 mice) and *Pr-set7^d/d* (n = 5 mice) uteri on PND5, PND10, and PND15. Data are presented as mean ± SEM. Two-tailed unpaired Student's *t*-test. ***p = 0.0001 (PND10), ***p = 4e-15 (PND15). **k** TUNEL analysis of the *Pr-set7^f/f* and *Pr-set7^d/d* uteri on PND5, PND10 and PND15. Dash lines represent the boundary between the epithelium and the stroma, as well as the stroma and the myometrium. Scale bar: 100 μm. **l** Number of TUNEL+ stromal cells per section in the *Pr-set7^f/f* (n = 5 mice) and *Pr-set7^d/d* (n = 5 mice) uteri on PND5, PND10 and PND15. Data are presented as mean ± SEM. Two-tailed unpaired Student's *t*-test. ***p = 4e-6 (PND5), ***p = 0.0002 (PND10), ***p = 5e-11 (PND15). Source data are provided as a Source Data file.

$p < 0.05$) were identified on PND5 and PND10, respectively (Fig. 5b). Specifically, 838 genes were upregulated and 509 genes were downregulated in the *Pr-set7^d/d* uterus on PND5. On PND10, 1084 genes were upregulated, while 1282 genes were downregulated. GO enrichment analysis revealed that genes upregulated upon *Pr-set7* deletion were mainly enriched in terms closely related to the activation of innate immune responses, including the response to virus, the production of cytokines and leukocyte migration (Fig. 5c).

The expression of genes involved in the NF-kappaB signaling, which serves as a crucial regulator of innate immunity and promotes the secretion of pro-inflammatory cytokines and chemokines, was significantly increased in the *Pr-set7^d/d* uterus (Supplementary Fig. 6a). WB analysis further proved the activation of the NF-kappaB pathway resulting from PR-SET7 deficiency, as manifested by elevated level of phosphorylated p65 (Fig. 5d). In addition, an increase in the expression of genes involved in leukocyte chemotaxis was observed due to PR-SET7 loss (Supplementary Fig. 6a). Gene set enrichment analysis (GSEA) indicated activation of the chemokine signaling pathway (Fig. 5e). Furthermore, obvious infiltration and gathering of leukocytes, especially macrophages, were noticed in the stromal compartment of the *Pr-set7^d/d* uterus, as visualized by CD45 and F4/80 staining (Fig. 5f and Supplementary Fig. 6b), implying an inflammatory environment in the stromal area.

As aforementioned, uterine PR-SET7 deficiency resulted in an exaggerated immune response to viral infection (Fig. 5c). Once viral infection takes place, the innate immune system of the host urges certain pattern recognition receptors (PRRs) to recognize microbe-derived pathogen associated molecular patterns (PAMPs, conserved microbial structures) or endogenous damage associated molecular patterns (DAMPs, tissue damage markers). These PRRs include toll-like receptors (TLRs), nucleotide oligomerization domain (NOD)-like receptors (NLRs), RIG-I-like receptors (RLRs) and so on[28]. Indeed, gene sets relevant to anti-viral pathways, including TLR signaling, NLR signaling and cytosolic DNA-sensing signaling, were remarkably enriched among the upregulated genes in the *Pr-set7^d/d* uterus (Fig. 5g and Supplementary Fig. 6c), and qRT-PCR analysis further confirmed the elevated level of several TLRs (Fig. 5h).

Upon the recognition of viruses, PRRs activate downstream transcription factors, such as interferon regulatory factor 3 (IRF3) and 7 (IRF7), leading to the production of interferons, a group of cytokines responsible for anti-viral defense. By binding with specific receptors, interferons further drive the phosphorylation of signal transducer and activator of transcription 1 (STAT1) and 2 (STAT2), ultimately leading to the production of ISGs and the amplification of anti-viral responses[29]. We observed a prominent increase in the level of p-IRF3 and p-STAT1, demonstrating the activation of the interferon signaling pathway (Fig. 5i). In addition, the expressions of 56 ISGs, including interferon induced transmembrane protein 6 (*Ifitm6*), interferon regulatory factor 8 (*Irf8*), interferon stimulated gene 15 (*Isg15*), 2′–5′ oligoadenylate synthetase 2 (*Oas2*) and Z-DNA binding protein 1 (*Zbp1*),

were significantly increased (fold change >= 1.2, $p < 0.05$) upon *Pr-set7* deletion (Fig. 5j, k).

Since the abovementioned studies were implemented on the whole uterus, we subsequently aimed to investigate whether stromal cells experienced the hyperactivation of anti-viral defenses in the absence of PR-SET7. SCRINSHOT was applied to detect the localization of the upregulated TLRs and ISGs. In fact, significantly more signals for *Tlr7*, *Isg15* and *Zbp1* were observed in both the epithelial and inner stromal compartments (Fig. 5l), consolidating that inner stromal cell underwent excessive anti-viral immune activation in the absence of PR-SET7.

H4K20me1 has been widely reported to play an indispensable role in the repair of DNA damage[15,18]. Consistently, the ablation of *Pr-set7* also led to considerable accumulation of DNA damage in the uterus. The p53 signaling pathway, which is elicited in response to DNA damage, was significantly activated in the *Pr-set7^d/d* uterus (Supplementary Fig. 6d). Meanwhile, γH2A.X staining revealed more cells with DNA double strand breaks in the uterine stroma of *Pr-set7^d/d* mutants compared to control mice (Supplementary Fig. 6e, f), which was consistent with the results of the TUNEL assay (Fig. 3k, l).

## H4K20me1 restrained the expression of ISGs in both direct and indirect manners

According to published literature, H4K20me1 exerts either promotive or repressive effects on gene transcription[16,30–32]. To excavate the underlying cause of compromised inner stromal differentiation and overwhelming anti-viral responses upon *Pr-set7* ablation, uterine tissues were subjected to CUT&Tag for H4K20me1 on PND5 and PND10. H4K20me1 modifications were predominantly distributed at gene promoters and introns (Fig. 6a–c).

Motif analysis indicated that gene promoters with H4K20me1 modifications possessed binding motifs for homeobox A10 (HOXA10) (Supplementary Fig. 7a), a critical mesenchymal transcription factor involved in female reproductive tract patterning[33] and the specification of the inner stroma (Fig. 2d), suggesting the role of H4K20me1 in the transcriptional regulation of genes related to inner stromal development. Besides, H4K20me1 modifications were detected on the gene bodies of *Prdm1* and *Gli1* (Supplementary Fig. 7b), which were obviously downregulated in the *Pr-set7^d/d* inner stromal cells (Fig. 4i).

Meanwhile, the promoters occupied by H4K20me1 exhibited motifs of interferon regulatory factor 4 (IRF4), reticuloendotheliosis oncogene (REL, NF-kappaB subunit) and IRF3 with a high probability (Supplementary Fig. 7a), implying the potential participation of H4K20me1 in modulating the transcription of innate immune related genes. The genes upregulated upon PR-SET7 loss were overlapped with genes modified by H4K20me1, and 242 genes directly repressed by H4K20me1 were identified (Fig. 6d). These genes were then subjected to GO analysis, and many of these genes were found to be associated with response to virus, leukocyte migration and activation of immune response (Fig. 6e). Therefore, we rationally speculated that

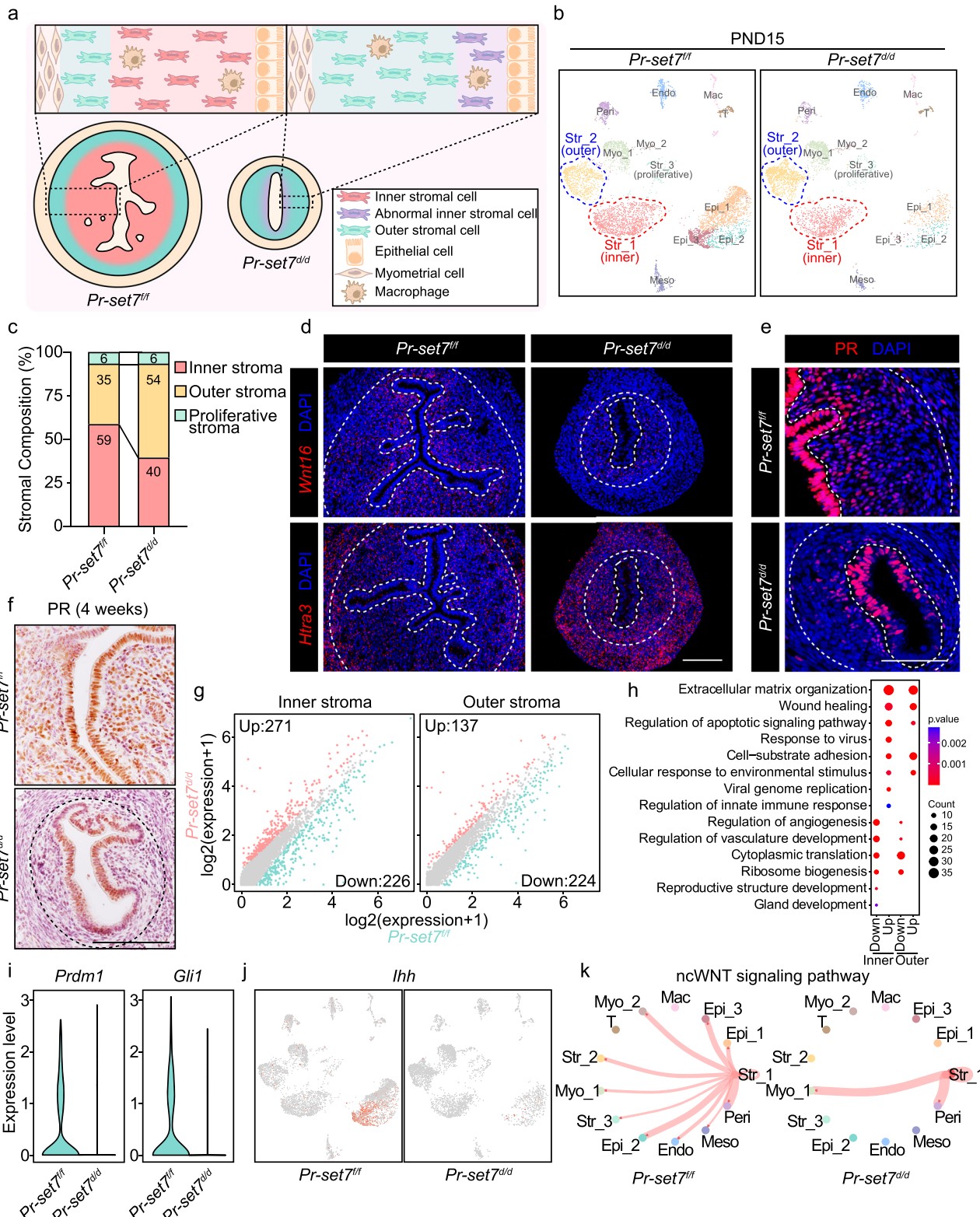

the upregulated ISGs upon *Pr-set7* deletion might be directly inhibited by H4K20me1 modifications. Indeed, combining CUT&Tag results with RNA-seq data, we observed that a portion of ISGs upregulated upon PR-SET7 deficiency were equipped with H4K20me1 modifications at promoters and gene bodies, including bone marrow stromal cell antigen 2 (*Bst2*), *Zbp1* and *Stat1* (Fig. 6f), suggesting that the expression of these genes was directly suppressed by H4K20me1. Nevertheless, we noticed that a subset of upregulated ISGs exhibited no obvious

H4K20me1 occupancy (Supplementary Fig. 7c), indicating an alternative mechanism by which H4K20me1 influenced the transcription of these genes.

As described above, pathways in response to virus were incited in the *Pr-set7*^d/d uterus. Given that viral infections rarely occur during postnatal uterine development, we hypothesized that the hyperactivated anti-viral reactions originated from the existence of ERVs. Under normal circumstances, ERVs are epigenetically silenced to

**Fig. 4 | The deficiency of PR-SET7 led to reduced number and derailed differentiation of inner stromal cells. a** Diagram illustrating uterine heterogeneity in *Pr-set7^f/f* and *Pr-set7^d/d* mice on PND15. **b** UMAP plot showing cell clusters in *Pr-set7^f/f* and *Pr-set7^d/d* uteri on PND15. **c** The percentage of stromal subsets in *Pr-set7^f/f* and *Pr-set7^d/d* uteri on PND15. **d** SCRINSHOT analysis of *Wnt16* and *Htra3* in the *Pr-set7^f/f* and *Pr-set7^d/d* uteri on PND15. Dash lines represent the boundary between the epithelium and the stroma, as well as the stroma and the myometrium. Scale bar: 100 μm. **e** IF analysis of PR in *Pr-set7^f/f* and *Pr-set7^d/d* uteri on PND15. Dash lines represent the boundary between the epithelium and the stroma, as well as the stroma and the myometrium. Scale bar: 100 μm. **f** IHC staining of PR in 4-week-old *Pr-set7^f/f* and *Pr-set7^d/d* uteri. Dash lines represent the boundary between the stroma and the myometrium. Scale bar: 100 μm. **g** Scatter plot showing the differentially expressed genes in the *Pr-set7^f/f* and *Pr-set7^d/d* inner and outer stroma. **h** GO functional analysis of the upregulated and downregulated genes in the inner and outer stromal cells due to PR-SET7 loss. Significance is based on over-representation analysis with clusterProfiler. **i** Violin plot indicating the expressions of *Prdm1* and *Gli1* in the *Pr-set7^f/f* and *Pr-set7^d/d* inner stroma. **j** UMAP visualization indicating the expression of *Ihh* in *Pr-set7^f/f* and *Pr-set7^d/d* uteri. **k** CellChat analysis showing the ncWNT signaling pathway among different cell clusters in *Pr-set7^f/f* and *Pr-set7^d/d* uteri.

maintain and ensure genome stability. The mechanisms by which ERVs are repressed include histone methylation (e.g., H3K9me3 and H4K20me3), DNA methylation and RNA methylation[34]. In order to verify that H4K20me1 was also involved in the silencing of ERVs, RNA-seq data were subjected to transposable element analysis, and a number of ERVs, including MER70-int:ERVL:LTR, were abnormally elevated upon PR-SET7 abrogation (Fig. 6g, h), which was also confirmed by qRT-PCR (Fig. 6i). Furthermore, CUT&Tag revealed H4K20me1 deposition on the gene body of MER70 as well as several other ERVs (e.g., LTR105_Mam:ERVL:LTR and MER68-int:ERVL:LTR) (Fig. 6j, k and Supplementary Fig. 7d).

Given the potential difficulty of short read sequencing in handling repetitive sequences, we noticed that the lengths of most ERVs we studied were shorter than 300 bp (Supplementary Fig. 8a), and different copies of the same ERV family are highly variable in sequences due to insertions, deletions and substitutions during evolution[35]. Since we performed paired-end 150 bp sequencing, the alignment of ERV sequences should be accurate. To further confirm this, we selected several ERV copies that were upregulated upon PR-SET7 loss according to our RNA-seq data (Supplementary Fig. 8b), amplified the full length of each copy from uterine stromal cell transcripts and performed Sanger Sequencing. The amplified full-length ERV copies were accurately mapped to ERV sequences downloaded from the RepeatMasker of the UCSC Genome Browser (Supplementary Fig. 8c), verifying the accuracy of our reads mapping.

In conclusion, these findings suggested that H4K20me1 directly restrained the transcription of certain ISGs; on the other hand, H4K20me1 indirectly prevents uncontrolled anti-viral immunity via mediating the silencing of ERVs.

## Declined H4K20me1 level led to viral mimicry responses and culminated in ZBP1-mediated necroptosis and apoptosis

Considering that the neonatal uterus is comprised of multiple cell types, in order to reinforce the significance of PR-SET7 specifically in stromal cells, primary mouse endometrial stromal cells isolated from the postnatal uterus were cultured in vitro and treated with DMSO (served as the control) or UNC0379, a selective substrate-competitive inhibitor of PR-SET7. A dose of 5 μM UNC0379 was sufficient to dampen H4K20me1 level in cultured uterine stromal cells (Fig. 7a). Consistent with the in vivo evidence, we observed aberrantly elevated levels of ERVs (Fig. 7b). Since ERVs were previously reported to cause the accumulation of dsRNAs[36], we performed dot blotting and IF analysis using the J2 antibody, which specifically recognizes dsRNAs. As expected, impairing PR-SET7 mediated H4K20me1 gave rise to a prominent increase in cytosolic dsRNA accumulation (Fig. 7c, d).

In the meanwhile, a viral mimicry cascade was ignited, as indicated by elevated p-IRF3 and p-STAT1 (Fig. 7e) as well as increased levels of ISGs, including *Bst2*, *Isg15*, *Oas2*, *Zbp1*, interferon induced transmembrane protein 3 (*Ifitm3*) and 2′−5′ oligoadenylate synthetase-like 2 (*Oasl2*) (Fig. 7f). Moreover, annexin V-FITC/PI co-staining evinced a greater ratio of cell death upon UNC0379 treatment (Fig. 7g and Supplementary Fig. 9). To further interrogate the cell death modality resulting from PR-SET7 inhibition, we examined markers of apoptosis (Fig. 7h), pyroptosis (Fig. 7i) and necroptosis (Fig. 7j), and found marked activation of both the apoptotic and necroptotic pathways, as revealed by elevated levels of cleaved CASPASE-3, p-RIPK3 (receptor-interacting protein kinase 3) and p-MLKL (mixed lineage kinase domain-like) (Fig. 7h, j).

In addition, to further consolidate our findings, we obtained primary uterine stromal cells from *Pr-set7^f/f* mice and performed transient transfection with plasmids carrying the Cre recombinase. Despite the relatively low transfection efficiency in primary uterine stromal cells (Supplementary Fig. 10a−c), we also observed the activation of interferon responses and cell death (Supplementary Fig. 10d, e).

In the subsequent study, we intended to bridge the gap between the hyperactivated immune response and cell death, and eventually put emphasis on ZBP1. ZBP1, the expression of which is induced by the interferon signaling, is capable of detecting the left-handed conformation of nucleic acids (referred to as Z-form nucleic acids) either produced during viral infection or transcribed from ERVs through its Zα domains and recruiting RIPK3 via shared receptor-interacting protein homotypic interaction motifs (RHIMs), which culminates in diverse modes of cell death including apoptosis and necroptosis[37]. Declined H4K20me1 by either *Pr-set7* deletion or UNC0379 treatment significantly upregulated the mRNA and protein levels of ZBP1 (Figs. 5j, l, and 7f, k). CUT&Tag revealed H4K20me1 occupancy at the promoter region of *Zbp1* (Fig. 6f), and chromatin immunoprecipitation (ChIP)-qRT-PCR using a pair of primers targeting the promoter locus of *Zbp1* further proved that *Zbp1* was a direct target of H4K20me1 (Fig. 7l). To explore whether the increase in cell death upon UNC0379 administration was mediated by ZBP1, endometrial stromal cells obtained from the uteri of *Zbp1^+/−* and *Zbp1^−/−* mice were subjected to UNC0379 treatment. According to annexin V-FITC/PI co-staining and immunoblotting, cell apoptosis and necroptosis caused by PR-SET7 inhibition were remarkably attenuated by ZBP1 abrogation (Fig. 7m−o).

These findings highlighted the pivotal role of ZBP1 in the execution of cell death caused by abnormally activated ERVs and viral mimicry responses upon PR-SET7 inhibition.

## Discussion

The spatiotemporal progression and potential regulatory machinery of postnatal uterine stromal development has been a long-standing enigma attributed to the scarcity of advanced approaches and appropriate genetic models. Here, we portrayed the transcriptomic atlas of the neonatal uterus throughout the postnatal development period at single-cell resolution. In line with the previous report[10], two distinct stromal subpopulations were defined, and the spatial distribution, developmental trajectory, and potential functions of these two stromal subsets were dissected. Furthermore, utilizing mouse model with uterine specific deletion of *Pr-set7*, we demonstrated that PR-SET7-mediated H4K20me1 facilitated the proper development of the inner stromal subset via epigenetically limiting exaggerated interferon responses and consequent cell death (Fig. 8).

Accumulated evidence has reinforced the heterogeneity of the endometrial stroma/decidua in the adult uterus[38–41]. Through scRNA-seq, Saatcioglu et al. unraveled the heterogeneity of the neonatal uterine stroma, and hypothesized that a population of *Misr2^+* cells at

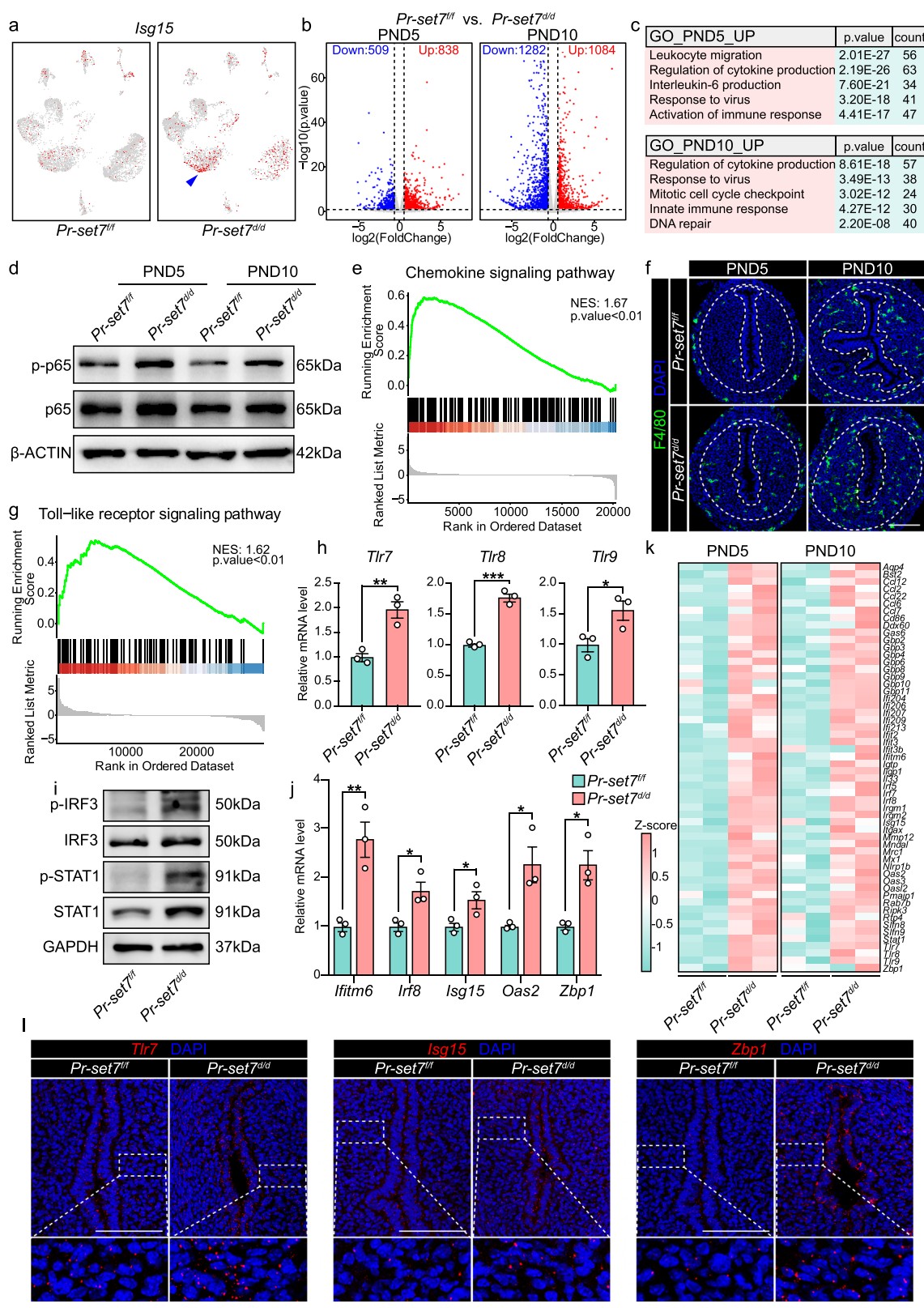

birth (before PND2) represented uterine stromal progenitors and gave rise to two stromal layers by PND6[10]. Since scRNA-seq was performed only on PND6 in this previous study, in order to provide a comprehensive and dynamic atlas of postnatal uterine stromal development, we carried out scRNA-seq analysis throughout the neonatal period (PND1-15). Our scRNA-seq data demonstrated that both stromal subsets already existed on PND1 in a less-differentiated state, and

underwent further differentiation in the first two weeks after birth. Therefore, we present an alternative view that the fates of uterine stromal cells are already determined at birth depending on their "position codes" along the luminal-myometrial radial axis. The previous study identified two stromal subpopulations and their distributions in the neonatal uterus, but the characteristics of these two distinct subsets remained elusive. In order to fill this gap, we described

**Fig. 5 | The uterus experienced overwhelming anti-viral reaction upon *Pr-set7* deletion. a** UMAP visualization of *Isg15* in the *Pr-set7*^f/f and *Pr-set7*^d/d uteri on PND15. The arrowhead indicates the expression of *Isg15* in the inner stroma of the *Pr-set7*^d/d uterus. **b** Volcano plot indicating the differentially expressed genes in the uteri after *Pr-set7* deletion on PND5 and PND10. Significance is based on the negative binomial test with DESeq2. Blue and red dots represent downregulated and upregulated genes, respectively, while genes that were not significantly differentially expressed are shown in gray. **c** GO functional analysis of the upregulated genes due to PR-SET7 loss. Significance is based on over-representation analysis with clusterProfiler. **d** WB analysis of p-p65 and t-p65 level in *Pr-set7*^f/f and *Pr-set7*^d/d uteri on PND5 and PND10. β-ACTIN served as a loading control. **e** GSEA analysis indicating the enrichment of chemokine signaling pathway related genes in the *Pr-set7*^d/d uteri compared with the *Pr-set7*^f/f uteri. Significance is based on functional class scoring with clusterProfiler. NES, normalized enrichment score. **f** IF analysis of F4/80 in *Pr-set7*^f/f and *Pr-set7*^d/d uteri on PND5 and PND10. Dash lines represent the boundary between the epithelium and the stroma, as well as the stroma and the myometrium. Scale bar: 100 μm. **g** GSEA analysis assessing the enrichment of toll-like receptor signaling pathway associated genes in the *Pr-set7*^d/d uteri compared with the *Pr-set7*^f/f uteri. Significance is based on functional class scoring with clusterProfiler. NES, normalized enrichment score. **h** QRT-PCR analysis of *Tlr7*, *Tlr8* and *Tlr9* mRNA level in *Pr-set7*^f/f (n = 3 mice) and *Pr-set7*^d/d (n = 3 mice) uteri on PND5. The values were normalized to *Gapdh* level. Data are presented as mean ± SEM. Two-tailed unpaired Student's *t*-test. **$p = 0.0063$ (*Tlr7*), ***$p = 0.0004$ (*Tlr8*), *$p = 0.0421$ (*Tlr9*). **i** Immunoblotting analysis of p-IRF3, IRF3, p-STAT1 and STAT1 in *Pr-set7*^f/f and *Pr-set7*^d/d uteri. GAPDH was used as a loading control. **j** QRT-PCR analysis of ISGs in *Pr-set7*^f/f (n = 3 mice) and *Pr-set7*^d/d (n = 3 mice) uteri on PND5. The values were normalized to *Gapdh* level. Data are presented as mean ± SEM. Two-tailed unpaired Student's *t*-test. **$p = 0.0085$ (*Ifitm6*), *$p = 0.0254$ (*Irf8*), *$p = 0.0461$ (*Isg15*), *$p = 0.0252$ (*Oas2*), *$p = 0.0139$ (*Zbp1*). **k** Heatmap showing the differentially expressed ISGs between *Pr-set7*^f/f and *Pr-set7*^d/d groups on PND5 and PND10. **l** SCRINSHOT analysis of *Tlr7*, *Isg15* and *Zbp1* in *Pr-set7*^f/f and *Pr-set7*^d/d uteri. Magnified images represent the SCRINSHOT signals in the stromal area. Scale bar: 100 μm. Source data are provided as a Source Data file.

the developmental trajectory of stromal subpopulations, transcription factors that potentially directed stromal differentiation, as well as the unique features of these two stromal subsets.

We observed an inequality of PR expression between the inner and outer stroma. The human endometrium is composed of the functional layer that experiences extensive proliferation, differentiation and decidualization in response to ovarian steroid hormones in each menstrual cycle, as well as the basal layer that remains relatively quiescent across the cycle[42]. Similarly in mice, stromal cells surrounding the implanting embryo transform into decidual cells under the influence of progesterone during pregnancy, leaving a thin layer of undifferentiated stroma adjacent to the myometrium[43]. Therefore, the relatively higher expression level of PR in the inner stroma implies its superior responsiveness to progesterone, which resembles the functional layer in humans, while the outer stroma is more analogous to the basal layer. Whether the inner and outer stroma in the neonatal uterus indeed represent the functional and basal layers that display diverging routes during the menstrual cycle/ pregnancy due to differences in progesterone responsiveness deserves further exploration.

Postnatal uterine development requires the intimate crosstalk between the epithelium and the mesenchyme/stroma[4]. Tissue recombination experiments clearly indicated that the uterine mesenchyme/stroma directed and specified epithelial proliferation and differentiation, while the epithelium was necessitated to support stromal and myometrial development[44–46]. For instance, mice with deletion of *Wnt7a*, which is specifically expressed in uterine epithelial cells, not only failed in adenogenesis, but also exhibited severe stromal defects[6]. Meanwhile, uterine stromal WNT5A has been demonstrated to be essential for the formation of glands[27]. Our scRNA-seq analysis revealed that the inner stromal cells, which were found to be in close proximity to the luminal epithelium, highly expressed genes related to gland morphogenesis. The expression level of these adenogenesis-associated genes peaked on PND10, when the glandular buds experienced extensive branching and invasion into the stroma, which underscored the significance of stromal-epithelial interactions in the development of uterine glands. CellChat analysis further revealed that the WNT and BMP signaling pathways potentially mediated the communications between inner stromal cells and glandular epithelial cells.

Although the previous study proved that the failure of uterine adenogenesis upon *Pr-set7* deletion resulted from limited epithelial population growth[26], it cannot be excluded that the dysfunctional inner stroma also contributed to compromised gland development, and/or that certain detrimental factors were emitted from the dying inner stromal cells, hampering the survival and maintenance of epithelial cells. On the other hand, the derailed differentiation and severe cell death of inner stromal cells in the *Pr-set7*^d/d uterus could partially arise from impaired epithelial functions and aberrant epithelial loss.

ERVs are evolutionary remnants originating from ancient retroviral infections during which the genetic materials of the viruses were integrated into the genome of host germ cells and transmitted across generations[34]. Despite the long-held misconception that ERVs are "junk DNA" without biological significance, emerging evidence has acknowledged their crucial contributions to various developmental events, such as embryogenesis and placentation, via functioning as regulatory elements (promoters or enhancers) and rewiring the transcriptional program[47–50]. However, aberrant transcription of ERVs disturbs genomic stability and is implicated in uncontrolled viral mimicry responses and even cell death[51]. Therefore, ERVs are tightly regulated by host surveillance mechanisms, including DNA methylation, histone modifications (e.g., H3K9me3 and H4K20me3) and RNA modifications (e.g., m6A), either individually or concurrently[34]. In addition to H3K9me3 and H4K20me3, a recent report demonstrated that PR-SET7-mediated H4K20me1 also participated in the transcriptional silencing of ERVs in trophoblast cells[20], and we discovered a similar epigenetic regulatory mechanism in the neonatal uterus in the present study. Further investigations are required to interrogate whether H4K20me1 mediates the repression of ERVs alone or via the interplay with other epigenetic regulatory machinery during postnatal uterine development.

ZBP1 is an ISG and serves as a cytoplasmic sensor of double-stranded nucleic acids to further activate the innate immune system against viral infections[52,53]. Previous studies have confirmed a critical role of ZBP1 in programmed cell death. Particularly, ZBP1 interacts with RIPK3 through the RHIM domains, resulting in the oligomerization and autophosphorylation of RIPK3, which drives parallel CASPASE-8-dependent apoptosis and MLKL-dependent necroptosis[54,55]. Additionally, ZBP1 also induces the ignition of pyroptosis by activating NLRP3[56]. Recently, the concept of PANoptosis, a multifaceted inflammatory cell death process involving the intricate interplay among apoptosis, necroptosis and pyroptosis, was proposed, and ZBP1 has been identified as a key upstream executor of PANoptosis[57]. Utilizing the primary culture of uterine stromal cells isolated from *Zbp1*^−/− mice and the administration of the PR-SET7 inhibitor, we proved that the inhibition of PR-SET7 led to massive apoptosis and necroptosis, which were largely restored by *Zbp1* deletion. However, the deficiency of ZBP1 alleviated but not eliminated cell death, suggesting the existence of alternative mechanisms contributing to the cell death induced by PR-SET7 inhibition.

Collectively, our study unearthed an unappreciated role of PR-SET7-mediated H4K20me1 in the regulation of postnatal uterine stromal subpopulation development via restricting hyperactivated interferon signaling and preventing uncontrolled cell death.

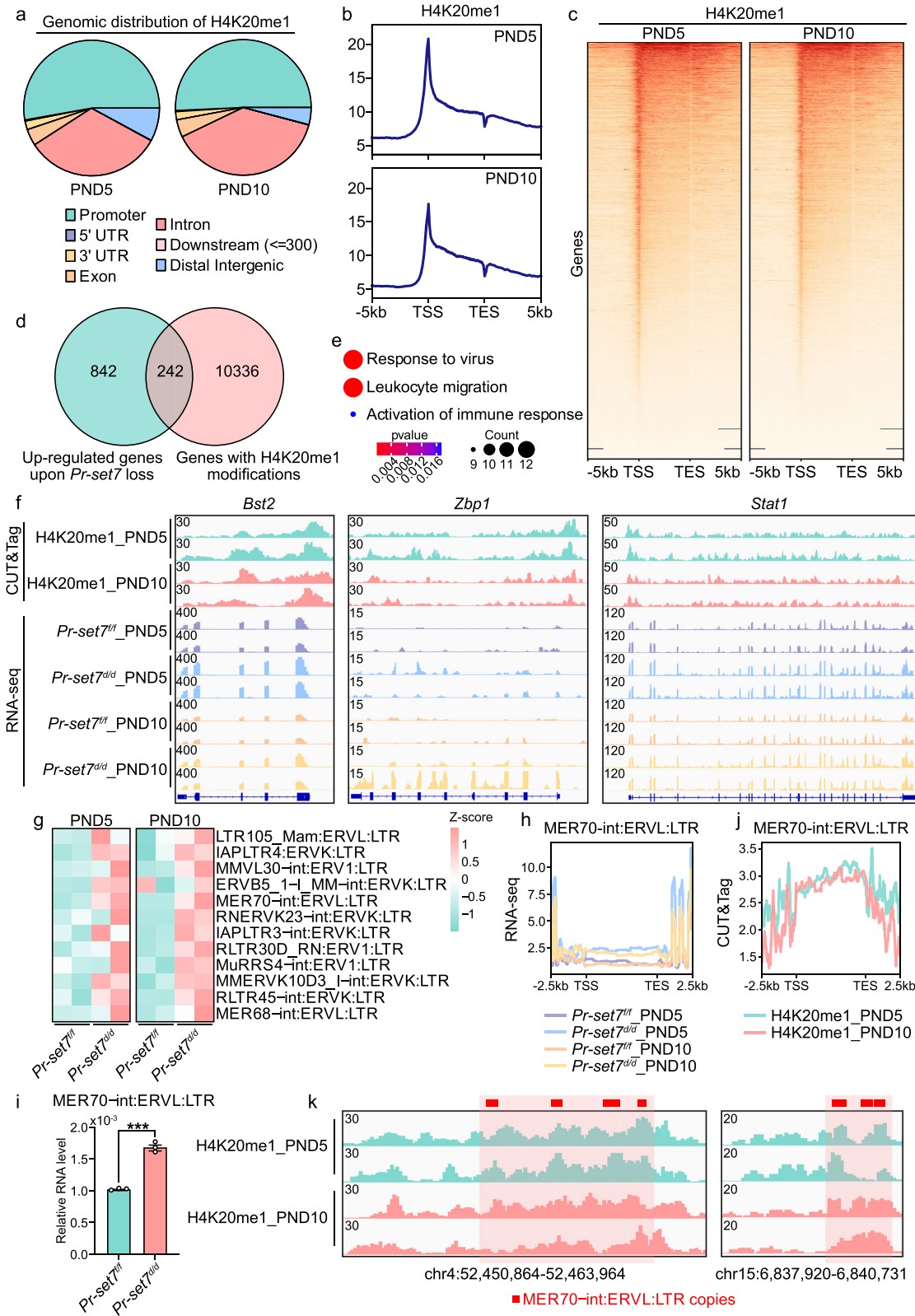

## Methods

### Animals

Neonatal (PND1-PND15) C57BL6 female mice were used in the present study. *Pr-set7 floxed* (*Pr-set7^{f/f}*), *Pgr-IRES-Cre,* and *Zbp1^{−/−}* transgenic mice used in the present study were constructed as previously described[24,58,59]. Mice with uterine specific deletion of *Pr-set7* (*Pr-set7^{d/d}*) were generated by crossing *Pr-set7^{f/f}* mice with *Pgr-IRES-Cre* mice. All

mice were housed in the animal care facility of Xiamen University (temperature: 22 ± 2 °C, humidity: 50–60%, 12 h light/dark cycle) according to the guidelines for the care and use of laboratory animals. All experimental procedures were approved by the Animal Care Committee of Xiamen University (XMULAC20170366). For the assessment of postnatal uterine development, *Pr-set7^{f/f}* and *Pr-set7^{d/d}* neonates were sacrificed, uteri of which were isolated and either frozen

**Fig. 6 | H4K20me1 repressed the expression of ISGs in both direct and indirect manners. a** Pie chart displaying the genomic distribution of H4K20me1. **b** Profile plot showing the distribution of normalized H4K20me1 CUT&Tag signals. TSS, transcription start site; TES, transcription end site. **c** Heatmap indicating the distribution of normalized H4K20me1 CUT&Tag signals. TSS, transcription start site; TES, transcription end site. **d** Venn diagram showing the upregulated genes upon PR-SET7 loss (n = 1084) and genes with H4K20me1 modifications (n = 10578). **e** GO enrichment analysis of genes that were directly suppressed by H4K20me1. Significance is based on over-representation analysis with clusterProfiler. **f** Genome browser view of normalized H4K20me1 CUT&Tag signals as well as *Pr-set7^f/f* and *Pr-set7^d/d* RNA-seq signals on a portion of ISGs that were upregulated upon *Pr-set7*

ablation and possessed H4K20me1 occupancy. **g** Heatmap indicating the abnormally upregulated ERVs due to PR-SET7 abrogation on PND5 and PND10. **h** Profile plot showing *Pr-set7^f/f* and *Pr-set7^d/d* RNA-seq signals on the body region of MER70-int:ERVL:LTR. TSS, transcription start site; TES, transcription end site. **i** QRT-PCR analysis of MER70-int:ERVL:LTR level in *Pr-set7^f/f* (n = 3 mice) and *Pr-set7^d/d* (n = 3 mice) uteri on PND5. The values were normalized to *Gapdh* level. Data are presented as mean +/- SEM. Two-tailed unpaired Student's *t*-test. ***p = 0.0002. **j** Profile plot showing normalized H4K20me1 CUT&Tag signals on the body region of MER70-int:ERVL:LTR. TSS, transcription start site; TES, transcription end site. **k** Genome browser view of normalized H4K20me1 CUT&Tag signals on MER70-int:ERVL:LTR copies. Source data are provided as a Source Data file.

in liquid nitrogen or fixed in 10% neutral buffered formalin overnight depending on subsequent experimental requirements.

### Primary uterine stromal cell culture

Uteri isolated from neonates were cut into small pieces and first digested using 25 mg/ml pancreatin (Sigma) and 6 mg/ml dispase (Gibco) for 10 min at 37 °C, and then incubated with 0.5 mg/ml collagenase (Sigma) for 30 min at 37 °C. After filtration, the obtained cells were plated and cultured in phenol red-free Dulbecco's Modified Eagle Medium/Nutrient Mixture F-12 Ham (Gibco) supplemented with 10% charcoal-stripped fetal bovine serum (Bioind). Adherent stromal cells were administrated with DMSO or 5 μM UNC0379 (MedChemExpress), or subjected to electrotransfection with plasmids carrying the CMV promoter-driven Cre recombinase.

### ISH

After removal from −80 °C, frozen sections were warmed at 37 °C for 5 min, and then fixed in 4% paraformaldehyde for 1 h at room temperature. Slides were incubated with Digoxigenin-labeled probes overnight at 60 °C. Following hybridization, uterine sections were incubated with Anti-Digoxigenin-Alkaline phosphatase overnight at 4 °C. Signals were detected with Nitrotetrazolium blue chloride/5-Bromo-4-chloro-3-indolyl phosphate. Images were captured using NIS-Elements D 4.50.00 (Nikon). Probe for *Pr-set7* was used in the present study, with detailed sequence information listed in Supplementary Table 1.

### Immunostaining

For fixed specimens, uterine tissue was dehydrated using graded ethanol, xylene and paraffin. Paraffin sections were deparaffinized, rehydrated, subjected to heat-mediated antigen retrieval before antigen blocking. For frozen sections and cell slides, they were fixed in 4% polyformaldehyde for 30 min at room temperature before blocking. Slides were blocked with 1% bovine serum albumin, incubated with primary antibodies overnight at 4 °C and then either horseradish peroxidase-labeled secondary antibodies (Zhongshan Golden Bridge Biotechnology, 1:200) or fluorescent dye-conjugated secondary antibodies (Jackson ImmunoResearch, 1:200) for 1 h at room temperature. Images were captured using NIS-Elements D 4.50.00 (Nikon) or Zen 2.3 (Zeiss). Primary antibodies against PR (CST, 8757, 1:200), H4K20me1 (ABclonal, A2370, 1:200), WT1 (Santa Cruz, sc-393498, 1:200), VIMENTIN (Abcam, ab92547, 1:200), α-SMA (BioGenex, MU128-UC, 1:200), PDGFRα (CST, 3174, 1:200), F4/80 (CST, 70076, 1:200), J2 (SCICONS, 10010500, 1:500), p21 (ABclonal, A19094, 1:200), Ki67 (Abcam, ab15580, 1:100), PCNA (Santa Cruz, sc-7907, 1:200), p-H3 (CST, 9701, 1:500), CD45 (CST, 70257, 1:200) and γH2A.X (CST, 9718, 1:500) were employed in this study, with detailed information listed in Supplementary Table 2.

### QRT-PCR

Total RNA was extracted from uterine tissue or cultured endometrial stromal cells using the TRIzol reagent (Invitrogen) according to the manufacturer's protocol. 1 μg RNA was reverse transcribed into cDNA. Expression level of target mRNAs was detected by quantitative real-

time PCR analysis with SYBR Green (TAKARA). All assays were performed at least three times. Primers used in this study were listed in Supplementary Table 1.

### WB

Proteins extracted from uterine tissue or cultured endometrial stromal cells were separated by sodium dodecyl sulfate polyacrylamide gel electrophoresis, and transferred to polyvinylidene difluoride membrane. The membrane with proteins was blocked with 5% skim milk for 1 h at room temperature, and incubated with primary antibodies overnight at 4 °C. In the subsequent procedure, the membrane was incubated with secondary antibodies (Zhongshan Golden Bridge Biotechnology, 1:5000) for 1 h at room temperature. Results were visualized using Supersignal West Pico (Thermo Scientific) according to the manufacturer's instructions. Primary antibodies against H4K20me1 (ABclonal, A2370, 1:1000), H3 (Abmart, P30266, 1:5000), p65 (CST, 8242, 1:1000), p-p65 (CST, 3033, 1:1000), β-ACTIN (Bioworld, AP0060, 1:5000), IRF3 (CST, 4302, 1:1000), p-IRF3 (CST, 79945, 1:1000), STAT1 (CST, 9172, 1:1000), p-STAT1 (CST, 9167, 1:1000), GAPDH (Bioworld, AP0063, 1:5000), J2 (SCICONS, 10010500, 1:5000), RIPK3 (CST, 15828, 1:1000), p-RIPK3 (CST, 91702, 1:1000), MLKL (CST, 37705, 1:1000), p-MLKL (CST, 37333, 1:1000), Cleaved CASPASE-3 (CST, 9661, 1:1000), GSDMD (Abcam, ab219800, 1:1000), ZBP1 (Santa Cruz, sc-271483, 1:500) and p21 (ABclonal, A19094, 1:1000) were applied in this study, with detailed information listed in Supplementary Table 2.

### SCRINSHOT

SCRINSHOT was performed as previously reported[60]. After removal from −80 °C, frozen sections were warmed at 45 °C for 3 min, and then fixed in 4% paraformaldehyde for 1 h at room temperature. After treated with the blocking solution (1× Ampligase Buffer, 0.05 M KCl, 20% Formamide deionized, 0.1 μM Oligo-dT, 0.2 μg/μl bovine serum albumin, 40 U/μl RNase Inhibitor, 10 μg/μl tRNA) for 30 min at room temperature, slides were incubated with specific Padlock probes for 20 min at 55 °C and then for 4 h at 45 °C. Following hybridization, uterine sections were subjected to probe ligation overnight at 25 °C using 0.5 U/μl SplintR (NEB, M0375) and rolling circle amplification overnight at 30 °C using 0.5 U/μl φ29 DNA polymerase (Vazyme, N106). Subsequently, sections were incubated with horseradish peroxidase-labeled detection oligos. Hybridization signals were amplified by the tyramide signal amplification system, and detected using Zeiss LSM 880 with Airyscan. Images were captured using Zen 2.3 (Zeiss). Padlock probes for *Wnt16*, *Htra3*, *Tlr7*, *Isg15* and *Zbp1* were used in the present study, with detailed sequence information listed in Supplementary Table 1.

### TUNEL

The TUNEL assay was performed using the One Step TUNEL Apoptosis Assay Kit (Beyotime, C1086) according to the manufacturer's protocol. Briefly, paraffin sections were deparaffinized, rehydrated and treated with 20 μg/ml proteinase K for 30 min at 37 °C, followed by the incubation of the TUNEL detection solution for 1 h at 37 °C in the dark.

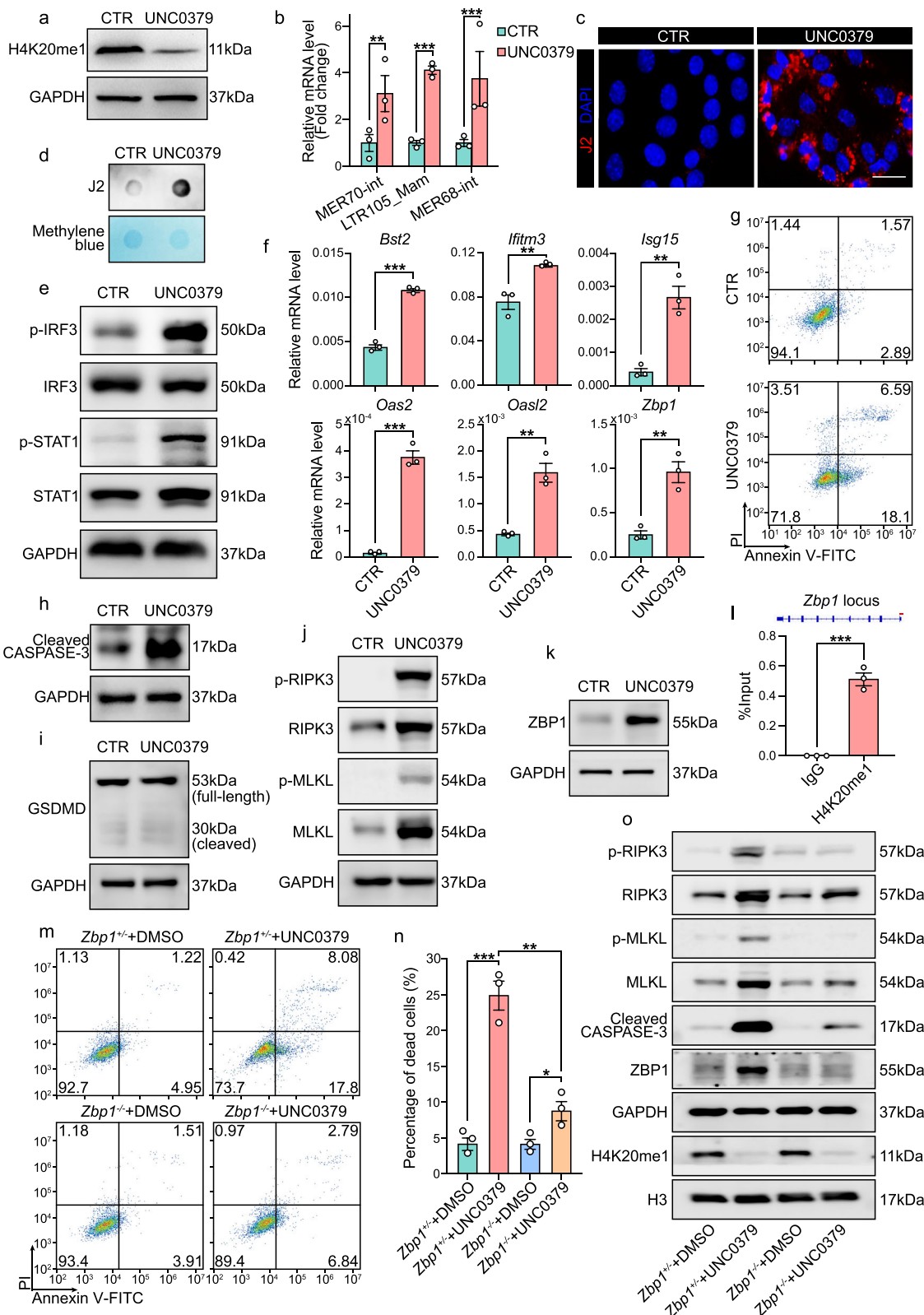

TUNEL signals were observed by Zeiss LSM 880 with Airyscan. Images were captured using Zen 2.3 (Zeiss).

## LacZ staining

Uteri of *Pgr-IRES-Cre;Rosa26-LacZ* neonates on PND1, 5, 10 and 15 were collected, fixed in 0.2% paraformaldehyde overnight at 4 °C and then dehydrated in 30% sucrose overnight at 4 °C. Dehydrated tissue was mounted in optimal cutting temperature compound for sectioning. Frozen sections were fixed in 0.2% paraformaldehyde for 10 min at 4 °C, and stained with the LacZ staining solution (0.1 M phosphate buffer pH 7.3, 2 mM $MgCl_2$, 0.01% sodium deoxycholate, 0.02% NP-40, 5 mM $K_3Fe(CN)_6$, 5 mM $K_4Fe(CN)_6$ and X-GAL) overnight at 37 °C. Images were captured using NIS-Elements D 4.50.00 (Nikon).

**Fig. 7 | Diminished H4K20me1 level led to viral mimicry responses and culminated in ZBP1-mediated apoptosis and necroptosis. a** WB analysis of H4K20me1 level in cultured endometrial stromal cells treated with DMSO (CTR) or UNC0379. GAPDH was used as a loading control. **b** QRT-PCR analysis of the mRNA level of ERVs in cultured uterine stromal cells treated with DMSO (CTR, n = 3 independent biological replicates) or UNC0379 (n = 3 independent biological replicates). The values were normalized to *Gapdh* level. Data are presented as mean ± SEM. Two-tailed unpaired Student's *t*-test. **\**p* = 0.0064 (MER70-int), \*\*\**p* = 0.0002 (LTR105_Mam), \*\*\**p* = 0.0008 (MER68-int). **c** IF analysis of J2 indicating dsRNA level in cultured endometrial stromal cells treated with DMSO (CTR) or UNC0379. Scale bar: 50 μm. **d** Dot blotting analysis of dsRNA level in cultured endometrial stromal cells treated with DMSO (CTR) or UNC0379. **e** Immunoblotting analysis of p-IRF3, IRF3, p-STAT1 and STAT1 in cultured endometrial stromal cells treated with DMSO (CTR) or UNC0379. GAPDH served as a loading control. **f** QRT-PCR analysis of the mRNA level of ISGs in cultured endometrial stromal cells treated with DMSO (CTR, n = 3 independent biological replicates) or UNC0379 (n = 3 independent biological replicates). The values were normalized to *Gapdh* level. Data are presented as mean ± SEM. Two-tailed unpaired Student's *t*-test. \*\*\**p* = 6e-5 (*Bst2*), \*\**p* = 0.0074 (*Ifitm3*), \*\**p* = 0.0032 (*Isg15*), \*\*\**p* = 0.0001 (*Oas2*), \*\**p* = 0.0029 (*Oasl2*), \*\**p* = 0.0050

(*Zbp1*). **g** Annexin V/PI apoptosis analysis of cultured endometrial stromal cells treated with DMSO (CTR) or UNC0379. **h–k** Immunoblotting analysis of Cleaved CASPASE-3 (**h**), GSDMD (**i**), p-RIPK3, RIPK3, p-MLKL and MLKL (**j**), as well as ZBP1 (**k**) in cultured endometrial stromal cells treated with DMSO (CTR) or UNC0379. GAPDH served as a loading control. **l** ChIP-qRT-PCR analysis of H4K20me1 on *Zbp1* locus. The data are representative of 3 independent biological replicates. Data are presented as mean ± SEM. Two-tailed unpaired Student's *t*-test. \*\*\**p* = 0.0003. **m** Annexin V/PI apoptosis analysis of cultured *Zbp1*+/− stromal cells treated with DMSO or UNC0379 and *Zbp1*−/− stromal cells treated with DMSO or UNC0379. **n** Percentage of cell death in *Zbp1*+/− stromal cells treated with DMSO (n = 3 independent biological replicates) or UNC0379 (n = 3 independent biological replicates) and *Zbp1*−/− stromal cells treated with DMSO (n = 3 independent biological replicates) or UNC0379 (n = 3 independent biological replicates). Data are presented as mean ± SEM. Two-tailed unpaired Student's *t*-test. \*\*\**p* = 0.0007, \*\**p* = 0.0023, \**p* = 0.0240. **o** Immunoblotting analysis of p-RIPK3, RIPK3, p-MLKL, MLKL, Cleaved CASPASE-3, ZBP1 and H4K20me1 in *Zbp1*+/− stromal cells treated with DMSO or UNC0379 and *Zbp1*−/− stromal cells treated with DMSO or UNC0379. GAPDH and H3 served as loading controls. Source data are provided as a Source Data file.

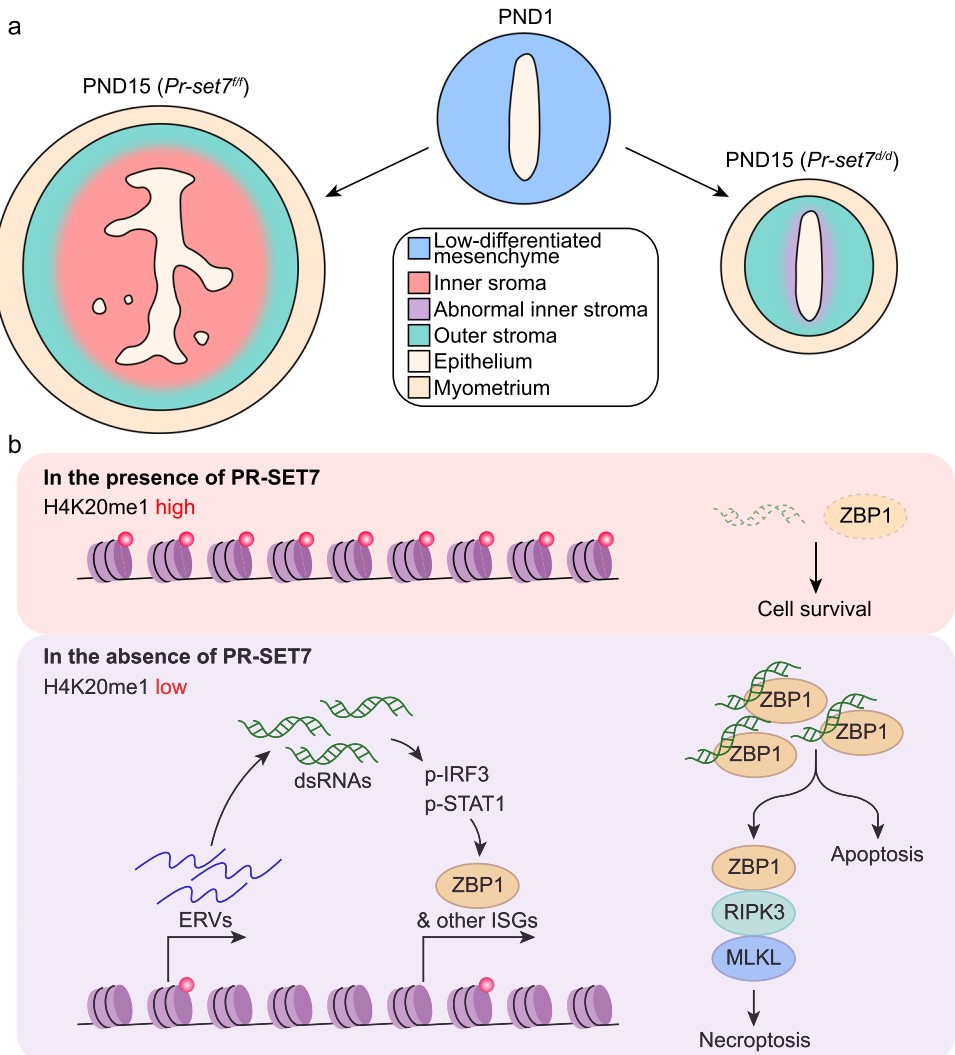

**Fig. 8 | Schematic diagram illustrating the role of PR-SET7-mediated H4K20me1 in postnatal uterine stromal development. a** Uterine *Pr-set7* deletion resulted in impaired inner stromal development after birth. **b** PR-SET7-mediated H4K20me1 epigenetically restricted exaggerated interferon responses via both direct and indirect manners, protecting stromal cells from ZBP1-mediated apoptosis and necroptosis.

## Dot blotting

1–2 µg total RNA extracted from cultured mouse endometrial stromal cells treated with DMSO or UNC0379 was dropped onto nitrocellulose membrane. Subsequently, the membrane was subjected to ultraviolet crosslinking, blocked with 5% skim milk for 1 h at room temperature and then incubated with primary antibody against J2 (SCICONS, 10010500, 1:5000) overnight at 4 °C. The next day, the membrane was incubated with secondary antibody (Zhongshan Golden Bridge Biotechnology, 1:5000) for 1 h at room temperature. Results were visualized using Supersignal West Pico (Thermo Scientific) according to the manufacturer's instructions. Detailed information of the J2 antibody was listed in Supplementary Table 2.

## Flow cytometry

Cultured mouse endometrial stromal cells treated with DMSO or UNC0379 were harvested and stained with Annexin V-FITC and propidium iodide (PI) in the dark for 20 min at room temperature according to manufacturer's instructions of the Annexin V-FITC Apoptosis Detection Kit (Beyotime, C1062). After staining, cells were immediately examined by flow cytometry using Beckman CytoFLEX S (CytExpert 2.4). Flow cytometry data were further analyzed by the FlowJo 10.8.1 software. Each experiment was repeated three times.

## RNA-seq

Total RNA was extracted from neonatal *Pr-set7*$^{f/f}$ and *Pr-set7*$^{d/d}$ uteri on PND5 and 10 using the TRIzol reagent (Invitrogen) according to the manufacturer's protocol. Purified RNA was subjected to RNA-seq using DNBSEQ-T7 (China, BGI).

## RNA-seq data analysis

RNA-seq raw data were filtered to obtain clean data after quality control by Trim Galore (0.6.4). High-quality clean data were aligned to the mouse reference genome (GRCm38) using STAR (2.7.3) with default parameters. Differentially expressed genes were identified using the DESeq2 (1.30.1) package with the criterion of fold change greater than 1.5 and $p$ value less than 0.05. The volcano plot and heatmap were performed using the ggplot2 (3.4.3) and pheatmap (1.0.12) package, respectively. The clusterProfiler (4.8.1) package was employed for GO and GSEA analysis.

## CUT&Tag

Uterine tissue isolated from neonates on PND5 and 10 was cut into small pieces and first digested using 25 mg/ml pancreatin (Sigma) and 6 mg/ml dispase (Gibco) for 10 min at 37 °C, and then incubated with 0.5 mg/ml collagenase (Sigma) for 30 min at 37 °C. After filtration, the collected cells were counted, and 100 thousand cells were subjected to subsequent CUT&Tag assay using the NovoNGS CUT&Tag 4.0 High-Sensitivity Kit (Novoprotein, N259-YH01) according to manufacturer's instructions. Briefly, cells were incubated with ConA magnetic beads for 10 min at room temperature, followed by primary antibody (anti-H4K20me1, ActiveMotif, 39727) incubation overnight at 4 °C, secondary antibody incubation for 1 h at room temperature, ChiTag transposome incubation for 1 h at room temperature, and then Tn5 tagmentation for 1 h at 37 °C. Next, DNA was extracted, amplified and purified. The constructed library was sequenced using Illumina NovaSeq 6000. Detailed information of the primary antibody used in the study was listed in Supplementary Table 2.

## CUT&Tag data analysis

CUT&Tag raw data were filtered to obtain clean data after quality control by Trim Galore (0.6.4). High-quality clean data were aligned to the mouse reference genome (mm10) using Bowtie2 (2.5.0). MACS2 (2.2.7.1) was applied for peak calling. Peak annotation was performed using the ChIPseeker (1.14.2) package. Heatmap and profile plot were obtained by deepTools (3.5.1).

## ChIP-qRT-PCR

Cultured mouse endometrial stromal cells were fixed with 1% formaldehyde for 10 min at room temperature, which was terminated by 0.125 M glycine for 10 min on ice. Fixed cells were collected and lysed in Lysis Buffer 1 (50 mM HEPES pH 7.5, 1 mM EDTA, 140 mM NaCl, 0.5% NP-40, 10% glycerol, 0.25% Triton X-100), Lysis Buffer 2 (10 mM Tris-HCl pH 8.0, 1 mM EDTA, 0.5 mM EGTA, 200 mM NaCl) and Lysis Buffer 3 (10 mM Tris-HCl pH 8.0, 1 mM EDTA, 0.5 mM EGTA, 100 mM NaCl, 0.1% Sodium Deoxycholate, 0.1% N-lauroylsarcosine). All lysis buffers were supplemented with protease inhibitors cocktail (Roche) before use. Chromatin DNA was sheared to 500–1000 bp using the BioRuptor sonicator (Diagenode) and incubated with primary antibody against H4K20me1 (ActiveMotif, 39727) bound to protein A magnetic beads (Invitrogen) overnight at 4 °C. After washing with Low Salt Buffer (50 mM HEPES pH 7.9, 2 mM EDTA, 150 mM NaCl, 1% Triton X-100, 0.5% Sodium Deoxycholate), High Salt Buffer (50 mM HEPES pH 7.9, 2 mM EDTA, 500 mM NaCl, 1% Triton X-100, 0.5% Sodium Deoxycholate), LiCl Buffer (10 mM Tris-HCl pH 8.0, 1 mM EDTA, 250 mM LiCl, 0.5% NP-40, 0.5% Sodium Deoxycholate), TE Buffer (10 mM Tris-HCl pH 8.0, 1 mM EDTA) and elution with Elution Buffer (100 mM NaHCO$_3$, 1% SDS), the protein-DNA complex was reversed overnight at 65 °C. Immunoprecipitated DNA and input DNA was purified and subjected to qRT-PCR. Detailed information of PCR primers and primary antibodies used in the study was listed in Supplementary Tables 1 and 2, respectively.

## Single-cell RNA-seq

Uteri of *Pr-set7*$^{f/f}$ neonates on PND1, 5, 10, 15 and *Pr-set7*$^{d/d}$ neonates on PND15 were isolated and digested into single-cell suspensions, which were then loaded into microfluidic devices using the Singleron Matrix Single Cell Processing System (Singleron). Subsequently, libraries were constructed according to the protocol of the GEXSCOPE Single Cell RNA Library Kit (Singleron). The constructed library was sequenced using Illumina NovaSeq 6000.

## Single-cell RNA-seq data analysis

Single-cell RNA-seq reads were processed by the CellRanger (4.0.0) pipeline with default and recommended parameters. The Seurat (4.3.0) toolkit was employed for quality control. High-quality cells (genes per cell >500 and the content of mitochondrial genes <5%) were subjected to subsequent analyses. 2000 highly variable genes were selected, and the top 50 principal components were calculated using the PCA function. The dimensionality reduction and cell clustering were performed using UMAP, and cell types were annotated according to well-known marker genes. Cell-cell interaction and communication analysis was performed using CellChat (1.5.0). For RNA velocity analysis, the BAM files were produced by default parameters of the CellRanger software. The velocyto.R package was used to calculate RNA velocity values for selected genes from each cell. Highly variably expressed genes computed by FindVariableFeatures of Seurat were further filtered out based on a cluster-wise expression, and the remaining highly variably expressed genes were selected as input for velocyto.R. Finally, The RNA velocity vectors were embedded in the UMAP plot.

## Statistical analysis

Statistical analysis was performed using GraphPad Prism 8.2.0 Software. All data were presented as mean ± SEM (standard error of mean). Each experiment included at least three independent samples. Comparisons between two groups were made by unpaired two-tailed Student's $t$-test. $P < 0.05$ was considered to indicate statistical significance.

## Statistics and reproducibility

All immunostaining, ISH, SCRINSHOT, WB, qRT-PCR, TUNEL, LacZ staining, dot blotting and flow cytometry experiments were repeated at least three times independently with similar results.

**Reporting summary**

Further information on research design is available in the Nature Portfolio Reporting Summary linked to this article.

## Data availability

The single-cell RNA-seq, bulk RNA-seq and CUT&Tag data generated in this study have been deposited in the National Center for Biotechnology Information Sequence Read Archive under accession code PRJNA1046685. Source data are provided with this paper.

## Code availability

Codes used in this study have been deposited at [https://github.com/dwb0211/PRSet7-scRNA].

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

## Acknowledgements

We are grateful to Professor Danny Reinberg (New York University School of Medicine) and Professor Wei Mo (School of Medicine, Zhejiang University) for providing *Pr-set7^{f/f}* and *Zbp1^{-/-}* transgenic mice. This work was supported by the National Key Research and Development Program of China (2022YFC2702500 and 2021YFC2700302 to H.W.), National Natural Science Foundation of China (82288102 and 82030040 to H.W., 82222026 to S.K., 82122026 and 32171117 to W.D., 82301886 to H.B.).

## Author contributions

H.B., N.D., Y.J., G.L., Y.G., X.L., Y.T., and H.C. performed experiments. H.W., W.D., and S.K. designed the study. H.B., Y.S., L.Z., J.L., and W.D. analyzed data. H.B., Y.S., N.D., H.W., and S.K. wrote the manuscript.

## Competing interests

The authors declare no competing interests.
