## [Peer Review File · Nature Communications]

PR-SET7 epigenetically restrains uterine interferon response and cell death governing proper postnatal stromal developmentREVIEWER COMMENTS

Reviewer #1 (Remarks to the Author):

The manuscript by Bao and coworkers investigates the role of PR-SET7 in uterine function. First this manuscript defines the cell populations of the developing prepubertal mouse uterus defining nicely the cell populations and potential signaling pathways between the compartments and cell clusters. The manuscript then investigates the role of Pr-set7 by ablating this gene in Pgr positive cells. Ablation of pr-set7 resulted in alterations of the cell populations of the uterine stroma with a decrease in the inner stroma population due to apoptosis. This manuscript then demonstrated that Pr-set7 mediated H4K20me3 repression of interferon signaling by silencing endogenous retroviral genes. This loss of retroviral silencing in Pr-set7 deleted mice resulted in ZBP1 mediated apoptosis. This manuscript uses single cell analysis and transcriptomic analysis to nicely dissect the mechanism of Pr-Set7 action and demonstrate the importance of silencing endogenous retroviruses in uterine physiology. All in all this is an outstanding mechanistic and novel work that will impact the field of developmental biology. The only issue that needs to be addressed is that using the Pgr IRES Cre will only ablate the PR-set7 in a subpopulation of cells. The impact on nonPgr+ cells mostly in the outer stroma must be discussed.

Reviewer #2 (Remarks to the Author):

The authors have examined the role of the histone methyltransferase PR-SET7 on uterine development and adenogenesis using a uterine-specific PR-SET7 knockout mouse and other experimental tools. They also characterize the different sub-populations of the uterine stroma/mesenchyme and epithelium in the normal and conditional knockout uterus, and these data are nice. They report that the loss of uterine epithelium and the infertility that they reported in a previous study is also accompanied by loss of the sub-epithelial stroma, and this demonstration of stromal cell loss is novel. They show data indicating that loss of PR-SET7 results in increased expression of a group of interferon-stimulated genes and the loss of silencing of endogenous retroviruses. Although a lot of data is presented here and some of it is totally new, significant aspects of this study merely confirm previously published work, and thus the amount of totally new and impactful information is modest. This limits enthusiasm for the current study, although interesting and valuable information is presented here.

Major problems

One major weakness of this study and the previous study this group published using this conditional knockout mouse (Cui et al, 2017) is that despite a lot of data suggesting apoptosis in the uterine mesenchyme/stroma at different times during postnatal development and the previous work showing epithelial apoptosis during this neonatal period, the authors do not present any data to indicate which cell types PR-SET7 is expressed in and when during the neonatal period it is expressed. Is it in both the uterine epithelium and stroma? When during the neonatal period is it present, or is it present all the time during uterine development? To really understand what PR-SET7 does in the uterus, this seems like essential information. Antibodies for PR-SET7 are available, so the authors could use immunohistochemistry to localize PR-SET7 expression if they wanted to. Alternatively, they could use Western blotting or PCR on their various cell populations from the neonatal uterus.

This group previously reported uterine epithelial apoptosis in conditional knockouts, but stromal-epithelial interactions are critical in the uterus, and it is possible that PR-SET7 could be normally expressed in the stroma, but that its loss led to epithelial apoptosis. Likewise, for the stromal apoptosis shown in conditional knockouts, is this a direct effect of loss of PR-SET7, or could loss of PR-SET7 in the epithelium lead to apoptosis in that cell layer, and then because the epithelium is needed for maintenance of the normal stroma, a secondary loss of stromal cells?

One focus of this current paper is the loss of sub-epithelial stroma, and in Fig. 3J and K, they present data to support this. However, in Fig. 3J, it looks like epithelial apoptosis is several fold higher than stromal apoptosis at PND 10 and 15. Therefore, is the loss of epithelium previous

reported the most important effect, as compared to the modest stromal apoptosis shown in this manuscript? In addition, the data in Fig. 4B also suggests the same thing... the cell type that is lost is preferentially the epithelial cells, at least at the PND 15 time point shown.

Another limitation of the present study is that a significant amount of the information presented here has been previously published by this group or other groups. For example, they describe the cKO PR-SET7 mouse here and describe the infertility and epithelial apoptosis in these mice, but this was also contained in their previous paper (Cui et al).

In this study, they describe the cell populations that can be sorted from the neonatal mouse stroma and epithelium, and that data is nice. However, the existence of more than one stromal population was previously described by Saatcioglu et al, eLIFE, 2019 (this is referenced by the authors).

They describe the existence of more than one epithelial cell population in the neonatal uterus, but this was previously described in greater detail by Spencer et al, PNAS, 2023 (which the authors do reference).

The data here showing that loss of PR-SET7 in the uterus results in increased expression of a group of interferon-stimulated genes and the loss of silencing of endogenous retroviruses is not totally surprising since similar data were published last year (Zhou et al, PNAS, 2023; ref. #20 in the manuscript) using trophoblast cells. In addition, Zhou showed increased apoptosis in their trophoblast cells following loss of PR-SET7, consistent with the apoptosis in uterine epithelium published previously and the stromal data in this manuscript. Therefore, is apoptosis following loss of PR-SET7 just a general phenomenon that would occur in many cells, or is this something unique in the uterine cell populations described here?

Another problem with this manuscript is that it is not sufficiently well written. The writing, structure of the paper and the English grammar all need significant work. The manuscript is definitely understandable, but the English problems affect the readability of the manuscript and detract from the impact of the work. While the difficulties of writing a manuscript in a language other than your native language are certainly acknowledged, improving the English in this manuscript would make it stronger. There are numerous commercial services that help with writing scientific papers and do so for a reasonable price; it is recommended that the authors utilize this type of service to improve both the quality of the English and overall readability of the manuscript.

Reviewer #3 (Remarks to the Author):

In this manuscript the authors characterize a novel function for PR-SET7 in the developing uterine stroma. Previous work analyzing Cre-mediated deletion of PR-SET7 in the developing uterus revealed a role for this protein in epithelial cells, where PR-SET7 results in accumulation of DNA damage and cell death. In this manuscript the authors extend this finding to uterine stromal cells where deletion of PR-SET7 similarly results in accumulation of DNA damage markers and cell death. In this manuscript the authors use single cell RNA seq to profile the cell types in the developing uterus. The authors then analyze the phenotype of mice with uterine deletion of PR-SET7 focusing on stromal cells. Deletion of PR-SET7 results in uterine stromal cell death accompanied by DNA damage, activation of the interferon pathway and viral mimicry.

The experiments in this paper appear to be performed well and the figures are clear and easy to follow. Overall, I feel this is a nice description of the cellular consequences of PR-SET7 deletion in the uterine stroma.

Comments:

1. I really have only one major comment regarding the experimental data, relating to the mechanism of viral mimicry triggering the interferon response. The activation of endogenous retroviruses (ERVs) presented in figure 6I appears to be extremely modest. The authors present increased levels of double stranded RNAs (dsRNAs) in figure 7B,C. Are readers intended to assume

that these are derived from ERV sequences? I find this proposed mechanism difficult to follow. Given the difficulty of short read sequencing in handling repetitive sequences, performing a direct measurement of ERV RNAs through an RNA northern blot would greatly strengthen the support for increased ERV expression.'

2. I found many issues relating to the writing of this paper.

There are a number of instances of odd or incorrect grammar throughout the manuscript, one example is found on lines 97-98:

"The highly heterogenous uterus was composed of various cell types, and experienced tremendous changes during development."

This sentence is written in the past tense, which is odd for the context it is use in, a suggested revision is:

"The highly heterogenous uterus is composed of various cell types, and undergoes tremendous changes during development."

I also found a number of instances of the use of overloaded and excessive descriptive language. For instance on lines 99-101:

"To achieve a comprehensive and profound understanding of the dynamic process of postnatal uterine development, uterine tissues were collected on PND1, 5, 10 and 15 and subjected to scRNA-seq (Fig. 1A)."

The word "profound" is excessive to describe an RNA-seq experiment.

Another example is found on lines 190-191:

"Uterine specific deletion of Pr-set7 hampered stromal growth due to robust cell death."

In biology the word "robust" generally refers to a process or trait that remains stable under perturbed/unstable conditions. In this case the authors seem to be using it to describe widespread cell death, so they should consider rewording this sentence.

There were also some examples of (perhaps unintentional) incorrect descriptions, for instance on lines 365-367:

"Meanwhile, γ H2A.X staining revealed enormous DNA double strand breaks in all cell types of the Pr-set7d/d uterus, including stromal cells (Fig. S6E-F)"

The word "enormous" as it is used here implies that the size of the DNA DSBs caused by Pr-set7 deletion is very large, however, the assay that the authors have used cannot make this conclusion. The authors are simply quantifying the number of cells in the stroma that show phospho-H2A.X immunoreactivity. This measurement cannot make any conclusion regarding either the number of DSBs per cell and certainly does not give any information about the size of the DSBs. A suggested revision is:

" γ H2A.X staining revealed more cells with DNA double strand in the uterine stroma of Pr-set7d/d mutants compared to control mice."

It is also important to note that γ H2A.X staining does not directly prove the presence of DSBs, a direct assay for DNA damage such as TUNEL or Comet assay would be required to definitively prove this.

I also noted instances of the use of informal/colloquial language, such as on lines 349-351:

"In addition, the expression of a bunch of ISGs was substantially increased upon Pr-set7 deletion, consisting of interferon induced transmembrane protein 6 (Ifitm6)"

Using "a bunch" to describe the number of differentially expressed genes is unacceptable in a scientific paper. The authors should provide a specific number and include the p-value and fold change thresholds that were used to define significantly differentially expressed genes.

These writing issues occur throughout the paper so I recommend that the authors spend time carefully reading through and correcting these problems.

My co-authors join me in expressing our sincere appreciation to the reviewers for their constructive and thoughtful comments which have led to significant improvement of the manuscript. We have performed additional experiments to strengthen our findings and incorporated new data into the revised manuscript (**Fig. 3a, Fig. 4g-h and Fig. 7b**). Besides, we have added two new supplementary figures in response to the reviewers' concerns (**Supplementary Fig. 8 and Fig. 10**). All changes in the revised manuscript have been highlighted in blue. Our POINT-BY-POINT RESPONSES are addressed below.

Reviewer #1 (Remarks to the Author):

The manuscript by Bao and coworkers investigates the role of PR-SET7 in uterine function. First this manuscript defines the cell populations of the developing prepubertal mouse uterus defining nicely the cell populations and potential signaling pathways between the compartments and cell clusters. The manuscript then investigates the role of Pr-set7 by ablating this gene in Pgr positive cells. Ablation of pr-set7 resulted in alterations of the cell populations of the uterine stroma with a decrease in the inner stroma population due to apoptosis. This manuscript then demonstrated that Pr-set7 mediated H4K20met repression of interferon signaling by silencing endogenous retroviral genes. This loss of retroviral silencing in Pr-set7 deleted mice resulted in ZPB1 mediated apoptosis. This manuscript uses single cell analysis and transcriptomic analysis to nicely dissect the mechanism of Pr-Set7 action and demonstrate the importance of silencing endogenous retroviruses in uterine physiology. All in all this is an outstanding mechanistic and novel work that will impact the field of developmental biology.

Response:

We sincerely appreciate the reviewer's positive feedback and constructive comment. We have made necessary revisions as suggested. Our responses to the reviewer's concern are addressed below.

The only issue that needs to be addressed is that using the Pgr Ires Cre will only ablate the PR-set7 in a subpopulation of cells. The impact on nonPgr+ cells mostly in the outer stroma must be discussed.

Response:

We sincerely appreciate the reviewer's constructive comment. As required, we have compared the differentially expressed genes upon *Pr-set7* loss between the inner (which is mostly composed of *Pgr*⁺ cells) and outer (which is mostly composed of *Pgr*⁻ cells) stroma. The numbers of down-regulated genes are comparable between the inner and outer stroma, while there are significantly fewer up-regulated genes in the outer stroma (**Revised Fig. 4g**). According to GO analysis, the down-regulated genes in both inner and outer stromal cells are enriched in biological processes related to angiogenesis and protein translation. Meanwhile, some down-regulated genes in the inner stromal cells are uniquely associated with reproductive structure development and gland development (**Revised Fig. 4h**). The up-regulated genes in both inner and outer stromal cells are related to extracellular matrix organization and wound healing, while many up-regulated genes in the inner stromal cells are particularly enriched in viral defense and innate immune response (**Revised Fig. 4h**). Consistently, most of the interferon stimulated genes up-regulated upon *Pr-set7* deficiency are observed in the inner stroma rather than the outer stroma (**Supplementary Fig. 5a**). These findings indicate that the deletion of *Pr-set7* in *Pgr*⁺ stromal cells (mostly in the inner stroma) also leads to changes in *Pgr*⁻ stromal cells (mostly in the outer stroma), potentially resulting from the altered uterine niche; however, the activation of anti-viral responses is peculiar to the inner stroma.

Reviewer #2 (Remarks to the Author):

The authors have examined the role of the histone methyltransferase PR-SET7 on uterine development and adenogenesis using a uterine-specific PR-SET7 knockout mouse and other experimental tools. They also characterize the different sub-populations of the uterine stroma/mesenchyme and epithelium in the normal and conditional knockout uterus, and these data are nice. They report that the loss of uterine epithelium and the infertility that they reported in a previous study is also accompanied by loss of the sub-epithelial stroma, and this demonstration of stromal cell loss is novel. They show data indicating that loss of PR-SET7 results in increased expression of a group of interferon-stimulated genes and the loss of silencing of endogenous retroviruses. Although a lot of data is presented here and some of it is totally new, significant aspects of this study merely confirm previously published work, and this the amount

of totally new and impactful information is modest. This limits enthusiasm for the current study, although interesting and valuable information is presented here.

Response:

We sincerely appreciate the constructive comments and valuable feedback provided by the reviewer. In regarding to the novelty of our study, we would like to elucidate our conceptual advances: (1) Through scRNA-seq analysis of the neonatal uterus from PND1 to PND15, we demonstrate that the cell fates of stromal subpopulations are already determined at birth, earlier than previously reported (PMID: 31232694). (2) We discover that unlike uterine epithelial cells that undergo apoptosis upon *Pr-set7* loss (PMID: 28731465), the ablation of *Pr-set7* in uterine stromal cells results in both apoptosis and necroptosis (a cell death modality distinct from apoptosis), indicating that different cell types in the uterus experience different cell death modes in response to PR-SET7 deficiency. (3) We prove that uterine stromal cell death upon *Pr-set7* loss is mediated by ZBP1, which is different from the TLR3-mediated necroptosis reported in the previous study using trophoblast cells (PMID: 37307441), suggesting that the necroptotic cell death caused by *Pr-set7* deletion may be mediated by different molecules and signaling pathways in different cell types. In addition, we have provided extra data and revised the manuscript in response to the reviewer's comments. Our detailed responses are addressed below.

Major problems

One major weakness of this study and the previous study this group published using this conditional knockout mouse (Cui et al, 2017) is that despite a lot of data suggesting apoptosis in the uterine mesenchyme/stroma at different times during postnatal development and the previous work showing epithelial apoptosis during this neonatal period, the authors do not present any data to indicate which cell types PR-SET7 is expressed in and when during the neonatal period it is expressed. Is it in both the uterine epithelium and stroma? When during the neonatal period is it present, or is it present all the time during uterine development? To really understand what PR-SET7 does in the uterus, this seems like essential information. Antibodies for PR-SET7 are available, so the authors could use immunohistochemistry to localize PR-SET7 expression if they wanted to. Alternatively, they could use Western blotting or PCR on their

various cell populations from the neonatal uterus.

Response:

We sincerely appreciate the reviewer's concern. Our previous work (PMID: 28731465) has revealed that PR-SET7 is ubiquitously expressed in all uterine cell types throughout the neonatal period (**Figure I, for reviewers**). Given that the anti-mouse PR-SET7 antibody applicable for immunohistochemistry is no longer available, we have performed in situ hybridization using a specific probe targeting *Pr-set7* to localize its expression. According to in situ hybridization results, *Pr-set7* is expressed in both uterine epithelium and stroma from PND1 to PND15 (**Revised Fig. 3a**), consistent with the previous study.

Figure I. Immunohistochemistry staining of PR-SET7 in the neonatal uterus (PMID: 28731465).

This group previously reported uterine epithelial apoptosis in conditional knockouts, but stromal-epithelial interactions are critical in the uterus, and it is possible that PR-SET7 could be normally expressed in the stroma, but that its loss led to epithelial apoptosis. Likewise, for the stromal apoptosis shown in conditional knockouts, is this a direct effect of loss of PR-SET7, or could loss of PR-SET7 in the epithelium lead to apoptosis in that cell layer, and then because the epithelium is needed for maintenance of the normal stroma, a secondary loss of stromal cells?

Response:

We appreciate the reviewer's concern. As illustrated above, PR-SET7 is expressed in both uterine epithelium and stroma during the neonatal period. Although we cannot exclude the possibility that the aberrant cell death in the uterine stroma causes epithelial loss due to hampered stromal-epithelial interactions and vice versa, which we have added in the discussion section in the revised manuscript (**the third paragraph, lines 549-557, highlighted in blue**), our previous study (PMID: 28731465) and current study have respectively demonstrated that cell death in both epithelial and stromal compartments are direct effects of PR-SET7 deficiency. In the previous study, Cui et al. found that *Pr-set7* knockdown in Ishikawa cells (human endometrial adenocarcinoma cell line) significantly increased the occurrence rate of apoptosis, suggesting that epithelial apoptosis upon *Pr-set7* loss is a cell-autonomous effect. Likewise, in order to reinforce the significance of PR-SET7 specifically in stromal cells, primary mouse uterine stromal cells were isolated and treated with UNC0379, a selective substrate-competitive inhibitor of PR-SET7. We proved that PR-SET7 inhibition led to viral mimicry responses and culminated in ZBP1-mediated necroptosis and apoptosis (**Figure 7**). In addition, to further consolidate our findings, we have obtained primary uterine stromal cells from *Pr-set7^{fl/fl}* mice and performed transient transfection with plasmids carrying the Cre recombinase. Despite the relatively low transfection efficiency in primary uterine stromal cells, we also observed the activation of interferon responses and cell death (**Revised Supplementary Fig. 10**). These results indicate that stromal cell death is directly attributed to PR-SET7 inhibition/deficiency.

One focus of this current paper is the loss of sub-epithelial stroma, and in Fig. 3J and K, they present data to support this. However, in Fig. 3J, it looks like epithelial apoptosis is several fold higher than stromal apoptosis at PND 10 and 15. Therefore, is the loss of epithelium previous reported the most important effect, as compared to the modest stromal apoptosis shown in this manuscript? In addition, the data in Fig. 4B also suggests the same thing... the cell type that is lost is preferentially the epithelial cells, at least at the PND 15 time point shown.

Response:

We appreciate the reviewer's concern. First of all, we would like to clarify that although epithelial cell death is more obvious on PND10, cell death in the inner stroma becomes severe on PND15 according to Fig. 3j (**Fig. 3k in the revised manuscript, also shown below**). These results indicate that cell death occurs earlier in the epithelium, which accounts for the more

significant loss of epithelial cells on PND15, as revealed by scRNA-seq in **Fig. 4b**.

With respect to the relatively earlier cell death in the epithelium, we have two potential explanations: (1) The ablation of *Pr-set7* by *Pgr-IRES-Cre* takes place earlier in the epithelium. According to scRNA-seq data, *Pgr* is detected at a low level in both stroma and epithelium on PND1. The level of *Pgr* is prominently increased in the epithelium on PND5, whereas its expression remains relatively low in the stroma until PND10 (**Figure II, for reviewers**). Therefore, massive cell death is first observed in epithelial cells on PND10, while inner stromal cell death only becomes obvious on PND15, five days later than that in the epithelium. (2) Since the cell death modes in uterine epithelial and stromal cells are different, the occurrence of cell death in these two cell types might be asynchronized.

Figure II. The expression pattern of *Pgr* in the neonatal uterus.

Another limitation of the present study is that a significant amount of the information presented here has been previously published by this group or other groups. For example, they describe the cKO PR-SET7 mouse here and describe the infertility and epithelial apoptosis in these mice, but this was also contained in their previous paper (Cui et al).

Response:

Thanks for the reviewer's concern. The major difference between our current work and the previous study by Cui et al. is that the present study mainly focuses on uterine stromal defects upon *Pr-set7* ablation rather than epithelial defects, while Cui et al. only explored the significance of PR-SET7 in uterine adenogenesis. We demonstrate that uterine *Pr-set7* deficiency leads to a reduced proportion of the inner stromal subset due to massive cell death induced by aberrant expression of ERVs, thus impeding uterine development. These findings are novel and have not been reported before.

In this study, they describe the cell populations that can be sorted from the neonatal mouse stroma and epithelium, and that data is nice. However, the existence of more than one stromal population was previously described by Saatcioglu et al, eLIFE, 2019 (this is referenced by the authors).

Response:

Thanks for the reviewer's concern very much. While we appreciate the study by Saatcioglu et al. and our scRNA-seq data further confirm the stromal subpopulations identified by them, there are two major conceptual advances of our study compared to the study by Saatcioglu et al.: (1) Saatcioglu et al. hypothesized that a population of *Misr2*⁺ cells at birth (before PND2) represented uterine stromal progenitors and gave rise to two stromal layers by PND6. Since scRNA-seq was performed only on PND6 in this previous study, in order to provide a comprehensive and dynamic atlas of postnatal uterine stromal development, we have carried out scRNA-seq analysis throughout the neonatal period (PND1-15). Our scRNA-seq data demonstrate that both stromal subsets already exist on PND1 in a less-differentiated state, and undergo further differentiation in the first two weeks after birth (**Fig. 1i-j**). Therefore, we present here an alternative view that the fates of uterine stromal cells are already determined at birth depending on their "position codes" along the luminal-myometrial radial axis. (2) The previous

study identified two stromal subpopulations and their distributions in the neonatal uterus, but the characteristics of these two distinct subsets remained elusive. In order to fill this gap, we have described the developmental trajectory of stromal subpopulations, transcription factors that potentially direct stromal differentiation, as well as the unique features of these two stromal subsets (**Fig. 2**). In order to emphasize the conceptual advances of our study, we have added the above statements in the discussion section in the revised manuscript (**the second paragraph, lines 501-518, highlighted in blue**).

They describe the existence of more than one epithelial cell population in the neonatal uterus, but this was previously described in greater detail by Spencer et al, PNAS, 2023 (which the authors do reference).

Response:

Thanks for the reviewer's concern. We really appreciate the study by Spencer et al., and the marker genes of epithelial cell populations identified by them are used for reference in our current study. However, the objectives of our work and the study by Spencer et al. are different. We mainly put emphasis on uterine stromal subpopulations during postnatal development, and therefore have not provided detailed descriptions of epithelial cell populations. Meanwhile, Spencer et al. performed scRNA-seq on isolated uterine epithelium of neonatal mice and only focused on uterine epithelial morphogenesis during the neonatal period.

The data here showing that loss of PR-SET7 in the uterus results in increased expression of a group of interferon-stimulated genes and the loss of silencing of endogenous retroviruses is not totally surprising since similar data were published last year (Zhou et al, PNAS, 2023; ref. #20 in the manuscript) using trophoblast cells. In addition, Zhou showed increased apoptosis in their trophoblast cells following loss of PET-SET7, consistent with the apoptosis in uterine epithelium published previously and the stromal data in this manuscript. Therefore, is apoptosis following loss of PET-SET7 just a general phenomenon that would occur in many cells, or is this something unique in the uterine cell populations described here?

Response:

We sincerely appreciate the reviewer's concern. In fact, cell death is a prevalent phenomenon upon *Pr-set7* loss; however, the modality of cell death may be discrepant under

different circumstances. For instances, *Pr-set7* null embryos display early embryonic lethality prior to the eight-cell stage and exhibit increased apoptosis (PMID: 19223465). Hepatocyte-specific deletion of *Pr-set7* results in necrosis (PMID: 25515659). The ablation of *Pr-set7* in trophoblast cells leads to overwhelming interferon response and necroptosis, a cell death mode distinct from apoptosis (PMID: 37307441). Our previous study by Cui et al. has revealed that uterine epithelial cells undergo apoptosis upon *Pr-set7* loss (PMID: 28731465). Nevertheless, in the present study, we unexpectedly discover that despite the obvious TUNEL (which detects both apoptosis and necroptosis) signals in *Pr-set7^{Δ/Δ}* stromal cells, only few cleaved CASPASE3 (which only indicates apoptosis) signals are observed (**Figure III, for reviewers**), implying that other cell death modality exists in the uterine stroma in response to *Pr-set7* loss. To further interrogate the cell death modality of uterine stromal cells, we have examined markers for apoptosis, pyroptosis and necroptosis. The activation of both apoptotic and necroptotic pathways has been proven (**Fig. 7h-j**).

Figure III. TUNEL and cleaved CASPASE-3 staining in the *Pr-set7^{Δ/Δ}* uteri on PND15.

With regards to the similar phenomenon published last year by Zhou et al. using trophoblast cells (PMID: 37307441), there are actually three major differences in the molecular mechanisms: (1) Zhou et al. reported that PR-SET7 mediated H4K20me1 indirectly repressed ISGs expression via silencing ERVs. However, our CUT&Tag data reveal that H4K20me1 either directly represses the transcription of ISGs or indirectly restricts the interferon response via silencing ERVs in uterine stromal cells, indicating a “double insurance mechanism” by which PR-SET7 safeguards postnatal stromal development. (2) We have identified previously unknown

target ERVs of PR-SET7-H4K20me1, including MER70-int, LTR105_Mam and MER68-int. (3) Zhou et al. demonstrated the involvement of TLR3 in necroptosis upon *Pr-set7* deficiency in trophoblast cells. Nevertheless, in our present study, we find that ZBP1 is significantly upregulated due to *Pr-set7* deletion. In addition, using primary uterine stromal cells obtained from *Zbp1*^{-/-} uteri, we prove that apoptosis and necroptosis upon *Pr-set7* loss in uterine stromal cells are mediated by ZBP1, suggesting that the cell death induced by *Pr-set7* deletion is mediated by different molecules and signaling pathways in different cell types.

Another problem with this manuscript is that it is not sufficiently well written. The writing, structure of the paper and the English grammar all need significant work. The manuscript is definitely understandable, but the English problems affect the readability of the manuscript and detract from the impact of the work. While the difficulties of writing a manuscript in a language other than your native language are certainly acknowledged, improving the English in this manuscript would make it stronger. There are numerous commercial services that help with writing scientific papers and do so for a reasonable price; it is recommended that the authors utilize this type of service to improve both the quality of the English and overall readability of the manuscript.

Response:

We sincerely appreciate the reviewer's concern to strengthen our manuscript. As suggested, we have performed language editing with the assistance of the Springer Nature Author Services (the editing certificate is provided, verification code: 2D3F-D78F-0DC6-9140-D767), and expect the revised manuscript to be satisfactory to the reviewer.

Reviewer #3 (Remarks to the Author):

In this manuscript the authors characterize a novel function for PR-SET7 in the developing uterine stroma. Previous work analyzing Cre-mediated deletion of PR-SET7 in the developing uterus revealed a role for this protein in epithelial cells, where PR-SET7 results in accumulation of DNA damage and cell death. In this manuscript the authors extend this finding to uterine stromal cells where deletion of PR-SET7 similarly results in accumulation of DNA damage markers and cell death. In this manuscript the authors use single cell RNA seq to profile the cell

types in the developing uterus. The authors then analyze the phenotype of mice with uterine deletion of PR-SET7 focusing on stromal cells. Deletion of PR-SET7 results in uterine stromal cell death accompanied by DNA damage, activation of the interferon pathway and viral mimicry.

The experiments in this paper appear to be performed well and the figures are clear and easy to follow. Overall, I feel this is a nice description of the cellular consequences of PR-SET7 deletion in the uterine stroma.

Response:

We sincerely appreciate the reviewer's comments and feedback regarding our manuscript. We have performed extra experiments to consolidate our findings and revised the manuscript as suggested. Our responses to the reviewer's concerns are addressed below.

Comments:

1. I really have only one major comment regarding the experimental data, relating to the mechanism of viral mimicry triggering the interferon response. The activation of endogenous retroviruses (ERVs) presented in figure 6I appears to be extremely modest. The authors present increased levels of double stranded RNAs (dsRNAs) in figure 7B,C. Are readers intended to assume that these are derived from ERV sequences? I find this proposed mechanism difficult to follow. Given the difficulty of short read sequencing in handling repetitive sequences, performing a direct measurement of ERV RNAs through an RNA northern blot would greatly strengthen the support for increased ERV expression.

Response:

We sincerely appreciate the reviewer's constructive comment. First, we would like to explain the relatively modest activation of ERVs presented in **Fig. 6i**. The qRT-PCR analysis was performed using the whole uterus composed of both PR^{positive} cells (in which *Pr-set7* was deleted) and PR^{negative} cells (in which *Pr-set7* was intact). Besides, even in PR^{positive} cells including epithelial cells, stromal cells and myometrial cells, the activation of interferon responses induced by ERVs was observed mainly in the inner stromal cells (**Supplementary Fig. 5a**). Meanwhile, the J2 dot blotting and immunofluorescence analyses presented in Fig. 7b-c (**Fig. 7c-d in the revised manuscript**) were performed in isolated uterine stromal cells, so the elevation of dsRNAs level was more obvious.

Considering the low abundance of ERV transcripts, it may be difficult to detect ERVs by northern blot. Alternatively, we have performed qRT-PCR to confirm the significantly elevated level of ERVs in uterine stromal cells upon PR-SET7 inhibition (**Revised Fig. 7b**). Regarding the question raised by the reviewer that it is difficult to handle repetitive sequences using short read sequencing, we find that the lengths of most ERVs in our study are shorter than 300bp (**Revised Supplementary Fig. 8a**), and different copies of the same ERV family are highly variable in sequences due to insertions, deletions and substitutions during evolution (PMID: 30962577). Since we performed paired-end 150bp sequencing, the alignment of ERV sequences should be accurate. To further confirm the accuracy of reads mapping, we select several ERV copies that are up-regulated upon *Pr-set7* loss according to our RNA-seq data (**Revised Supplementary Fig. 8b**), amplify the full length of each copy from uterine stromal cell transcripts and perform Sanger Sequencing. The amplified full-length ERV copies are accurately mapped to ERV sequences downloaded from the RepeatMasker of the UCSC Genome Browser (**Revised Supplementary Fig. 8c**).

2. I found many issues relating to the writing of this paper.

There are a number of instances of odd or incorrect grammar throughout the manuscript, one example is found on lines 97-98:

"The highly heterogenous uterus was composed of various cell types, and experienced tremendous changes during development."

This sentence is written in the past tense, which is odd for the context it is use in, a suggested revision is:

"The highly heterogenous uterus is composed of various cell types, and undergoes tremendous changes during development."

Response:

We apologize for the incorrect grammars and have corrected them in the revised manuscript (**Lines 94-95, 209, 216, 283, 314-315, 329, 333, 340, 343, 363, 371, 406, 408, 410, 437, 446, 467-468, 471 and 538**).

I also found a number of instances of the use of overloaded and excessive descriptive language.

For instance on lines 99-101:

"To achieve a comprehensive and profound understanding of the dynamic process of postnatal uterine development, uterine tissues were collected on PND1, 5, 10 and 15 and subjected to scRNA-seq (Fig. 1A)."

The word "profound" is excessive to describe an RNA-seq experiment.

Response:

Thanks for pointing out the problem. We have deleted this word in the revised manuscript (Lines 95-97).

Another example is found on lines 190-191:

"Uterine specific deletion of Pr-set7 hampered stromal growth due to robust cell death."

In biology the word "robust" generally refers to a process or trait that remains stable under perturbed/unstable conditions. In this case the authors seem to be using it to describe widespread cell death, so they should consider rewording this sentence.

Response:

We apologize for the misuse of words and have reworded the sentence as suggested in the revised manuscript (Lines 191-192).

There were also some examples of (perhaps unintentional) incorrect descriptions, for instance on lines 365-367:

"Meanwhile, γ H2A.X staining revealed enormous DNA double strand breaks in all cell types of the Pr-set7d/d uterus, including stromal cells (Fig. S6E-F)"

The word "enormous" as it is used here implies that the size of the DNA DSBs caused by Pr-set7 deletion is very large, however, the assay that the authors have used cannot make this conclusion. The authors are simply quantifying the number of cells in the stroma that show phospho-H2A.X immunoreactivity. This measurement cannot make any conclusion regarding either the number of DSBs per cell and certainly does not give any information about the size of the DSBs. A suggested revision is:

" γ H2A.X staining revealed more cells with DNA double strand in the uterine stroma of Pr-set7d/d mutants compared to control mice."

It is also important to note that γ H2A.X staining does not directly prove the presence of DSBs, a

direct assay for DNA damage such as TUNEL or Comet assay would be required to definitively prove this.

Response:

We sincerely appreciate the reviewer's concern and apologize for the incorrect descriptions. As suggested, we have revised the description (**Lines 364-366**). Moreover, the TUNEL assay also confirms the DNA damages in *Pr-set7^{d/d}* uteri (**Fig. 3k**).

I also noted instances of the use of informal/colloquial language, such as on lines 349-351: "In addition, the expression of a bunch of ISGs was substantially increased upon Pr-set7 deletion, consisting of interferon induced transmembrane protein 6 (Ifitm6)" Using "a bunch" to describe the number of differentially expressed genes is unacceptable in a scientific paper. The authors should provide a specific number and include the p-value and fold change thresholds that were used to define significantly differentially expressed genes.

Response:

We sincerely appreciate the reviewer's concern and apologize for the informal language. As required, we have provided the number, p-value and fold change thresholds that were used to define up-regulated ISGs (**Lines 347-352**).

These writing issues occur throughout the paper so I recommend that the authors spend time carefully reading through and correcting these problems.

Response:

We sincerely appreciate the reviewer's concern to strengthen our manuscript. We have read through the manuscript and corrected these mistakes as required. In addition, we have performed language editing with the assistance of the Springer Nature Author Services (the editing certificate is provided, verification code: 2D3F-D78F-0DC6-9140-D767), and expect the revised manuscript to be satisfactory to the reviewer.

REVIEWERS' COMMENTS

Reviewer #1 (Remarks to the Author):

The authors have addressed my concerns on this manuscript. The manuscript represents new exciting mechanistic data.

Reviewer #3 (Remarks to the Author):

The authors have addressed my concerns. The language and grammar of the manuscript have been greatly improved.